# Geo-historical database of flood impacts in Alpine catchments (HIFAVa database, Arve River, France, 1850 – 2015).

Eva Boisson[1], Bruno Wilhelm[1], Emmanuel Garnier[2], Alain Mélo[3], Sandrine Anquetin[1], Isabelle Ruin[1].

[1] Univ. Grenoble Alpes, CNRS, IRD, Grenoble INP, IGE, 38000 Grenoble, France.
[2] Univ. Franche-Comté, CNRS, LCE, 25000 Besançon, France.
[3] AXALP, Annecy, France; associate member at Univ. Savoie Mont-Blanc, CNRS, EDYTEM - UMR 5204, 73370 Le Bourget du Lac, France.

Correspondance to: Eva Boisson (eva.boisson@univ-grenoble-alpes.fr)

**Abstract**

In France, flooding is the most common and damaging natural hazard. Global warming is expected to exacerbate flood risk and could be more pronounced in the European Alps which are experiencing a high warming rate, likely to lead to heavier rainfall events. Alpine valleys are densely populated, potentially increasing exposure and vulnerability to flood hazard. The study of historical records is highly relevant to understand long-term flood occurrence and related socio-economic impacts in relation to changes in the flood risk components (i.e. hazard, exposure and vulnerability).

To this aim we introduce the newly constituted database of *Historical Impacts of Floods in the Arve Valley* (HIFAVa) located in French Northern Alps starting in 1850. This quite unique database reports historical impacts related to impact events occurrences in a well-documented Alpine catchment that encompasses both hydrological and socio-economical diversity.

After a complete description of the database (collection, content and structure), we explore the distribution of the recorded impacts with respect to their characteristics and evolution in both time and space. The analysis reveals that small mountain streams and particularly glacial streams caused more impacts (67%) than the main river. While an increase in heavy rainfall and ice melt are expected to enhance flood hazard in small Alpine catchments, this finding calls for greater attention to flood risk assessment and management in small catchments. The analysis also reveals an increasing occurrence of impacts from 1920 onwards, for which possible factors are discussed. Further work is, however, needed to conclude on the respective contribution of the source effect, the increase in flood hazard or the exposure of goods and people.

**Keywords:** flood risk, history, socio-economic impacts, exposure, vulnerability, French Alps.

# 1. Introduction.

On the mainland French territory flood is the most common and damaging natural hazard in terms of economic cost and number of municipalities concerned (Ministère de la Transition écologique, 2020). In highland regions, these events can be caused, among others, by summer thunderstorms, rain on thaw saturated soils, rain on snow or by glacial lake outburst (Merz and Blöschl, 2003). The topography induces flood events with highly contrasted dynamics; from sudden events with large sediment transport in the upstream small catchments to multi-day events flooding large parts of the valley floor. This diversity of hydrological dynamics adds to the complexity in flood risk management. Furthermore, climate change is expected to increase extreme precipitation (Min et al., 2011) that could in turn increase flooding (Gobiet et al., 2014; Blöschl et al., 2020). This is especially the case for the European Alps where an increase in summer heavy rainfalls (Giorgi et al., 2016; Ménégoz et al., 2020) may threaten densely populated mountainous valleys, which are especially vulnerable to climate extremes (IPCC, 2019). With its long history of flooding, the densely populated Arve valley located in the Northern French Alps is indeed prone to experience increased flood risk as a result of global warming in the future.

Historical records constitute a source of reliable data to characterize past hydrological events because they contribute to give a comprehensive representation of these events and of their changes over long time scales in spite of the lack of instrumental data (Garnier and Desarthe, 2013; Barriendos et al., 2014; Wetter, 2017; Macdonald and Sangster, 2017; Wilhelm et al., 2019). They also allow to apprehend changes in flood risk since they document how societies were impacted by past flooding events. Here, we consider impacts accordingly to the IPCC (2012) definition as all types of outcomes for humans, society and ecosystems occurring in the aftermath of a physical phenomenon, i.e., any disturbance, damage, casualties or destruction described in the historical archives and related to a flood event. The historical analysis of past events is useful for the study of catastrophe as we can hypothesize that these remarkable events are etched in the community's memory (Papagiannaki et al., 2013a). Indeed, it is because these events have impacted society that they are recorded in the historical records, i.e. have left a "social signature". Those high impact events can come close to the notion of a catastrophe, as they can lead to societal upheaval (Soanes and Stevenson, 2009) sometimes deleterious but also beneficial (behavioral change promoting prevention) (Garnier, 2017). High impact events are by nature rare, often resulting in a lack of available data (e.g. description of the event, time, extent, damages caused etc.). However, historical analysis allows a social and spatial-temporal contextualization of the data (Giacona et al., 2017), making the reconstruction (date, impacts) of major flood events possible (Barriendos et al., 2003, 2019) and attesting the social apprehension of the phenomenon (Gil-Guirado et al., 2016). Numerous historical databases were built to document past flood occurrence and magnitude, such as the Prediflood database (Barriendos et al., 2014), and some, as the database from Thoumas (2019), allow to analyze the climatic fluctuations. In contrast to these latter databases focusing on hydrological events, some databases gather information on the socioeconomic impacts of floods such as the APAT database (Lastoria et al., 2006), the press database on natural hazards and climate change from Llasat et al. (2009), the database of high-impact weather events in Greece from Papagiannaki et al. (2013a), the EUFF database (Petrucci et al., 2019) and the SMC-Flood database (Gil-Guirado et al., 2019). Some databases stand out as the participative flood database ORRION (Giacona et al., 2019), the on-line information resources of the Chronology of British Hydrological Events (Black and Law, 2004) or the database built by the RISC-KIT project (Garnier et al., 2018) which aim at developing methods, management approaches and explore trajectories of vulnerability.

To our knowledge there are no similar works published about flood impacts identified in the
archives over historical timescale (i.e. over longer time periods with only the last few decades
covered by instrumental data) and in a mountanous catchment. There are numerous
databases on flood impacts but most of them refer either to the instrumental period (Schlögl et
al., 2021) or to other hazard than flooding (Zgheib et al., 2020; Papagiannaki et al., 2013b;
Giacona et al., 2017) or cover a larger area (Barriendos et al., 2014; Macdonald and Sangster,
2017).
Floods, as natural hazards, are physical phenomena naturally occurring and can, when
certain conditions are met, cause harm to societies. They can be interpreted as a social
construction (Beck, 1992) since exposure of human activities and social vulnerability play a
large role in the severity of the impacts. Flood impacts databases, constituted from historical
records, can be considered as the expression of society's concerns, risk perceptions (fear,
habit) and values (based on reported impacts). The recording of flood impacts, or the failure
to record them, provides a subjective measure of the events that were considered worth
reporting for various reasons across historical periods. Flood impacts result from the
interaction between the natural phenomenon and the dynamics of exposure and vulnerability.
As vulnerability we understand the inclination to damage of various exposed goods, activities
or people constituting a given territory (Leone and Vinet, 2006). We consider the vulnerability
as a dynamic system articulated to numerous physical and societal factors (Antoine, 2011;
Terti et al., 2015). This system can evolve in time and space (Cutter, 2003). Major natural
disasters, such as floods, are often displayed as unforeseeable events whereas the historical
facts give evidence of the contrary (Garnier, 2016, 2019). Yet society's vulnerability may
increase as past disasters are forgotten, leading to a "society of risk" (Garnier, 2019). The
historical approach allows consideration and exploration of the trajectories of hazard and
vulnerability in response to changes in climate, land use and flood risk management (Gil-
Guirado et al., 2016).
The present paper introduces a newly constituted database of flood impacts of the Arve
River and its tributaries (Northern French Alps). The database called "*Historical Impacts of*
*Floods in the Arve Valley*" (HIFAVa) covers all impacts caused by hydrological events that
occurred since 1850.
The study of this database, probably the first one documenting flood impacts over historical
time scale in a mountainous catchment, ultimately aims at analyzing the interactions between
social and natural dynamics engendering these impacts. In this paper we present the dataset
and first results of the impacts analysis with respect to their nature and evolution in both time
and space.

## 2. Study area: the Arve River.

### 2.1.        Description of the physical setting of the Arve River.
The Arve River is located in the Northern French Alps (Figure 1), flowing from the high
elevations of the Mont-Blanc summit (4810m a.s.l.) to the Swiss lowlands (330m a.s.l.), where
it flows into the Rhône River. The surface area of its catchment is 2164 km² with the largest
part higher than 1000m a.s.l.. The main tributaries of the Arve River are the Giffre, the Borne,
the Menoge and the Foron Rivers.
Since 1850, i.e. the start date of the studied period, almost all the current diking systems were
already in place (Mougin, 1914; Gex, 1924; Peiry and Bravard, 1989; ACTHYS-Diffusion,
2017) and from 1880 onward, most of the dyke construction work was completed and their
nature did not significantly change after this date. As shown by the study of ACTHYS-Diffusion
(2017), most of the 21st century's development of the diking systems of the Arve and the Borne
Rivers protecting the city of Bonneville concern the construction of weirs to fight against
generalized stream incision due to the important extraction of materials in the rivers. Some
repair works were carried out on dikes during the 20th century and new works (development of
weirs, raising of dikes) only happened at the beginning of the 2000s. Unfortunately, this study
is limited to the area of Bonneville and has not been replicated to the rest of the territory.

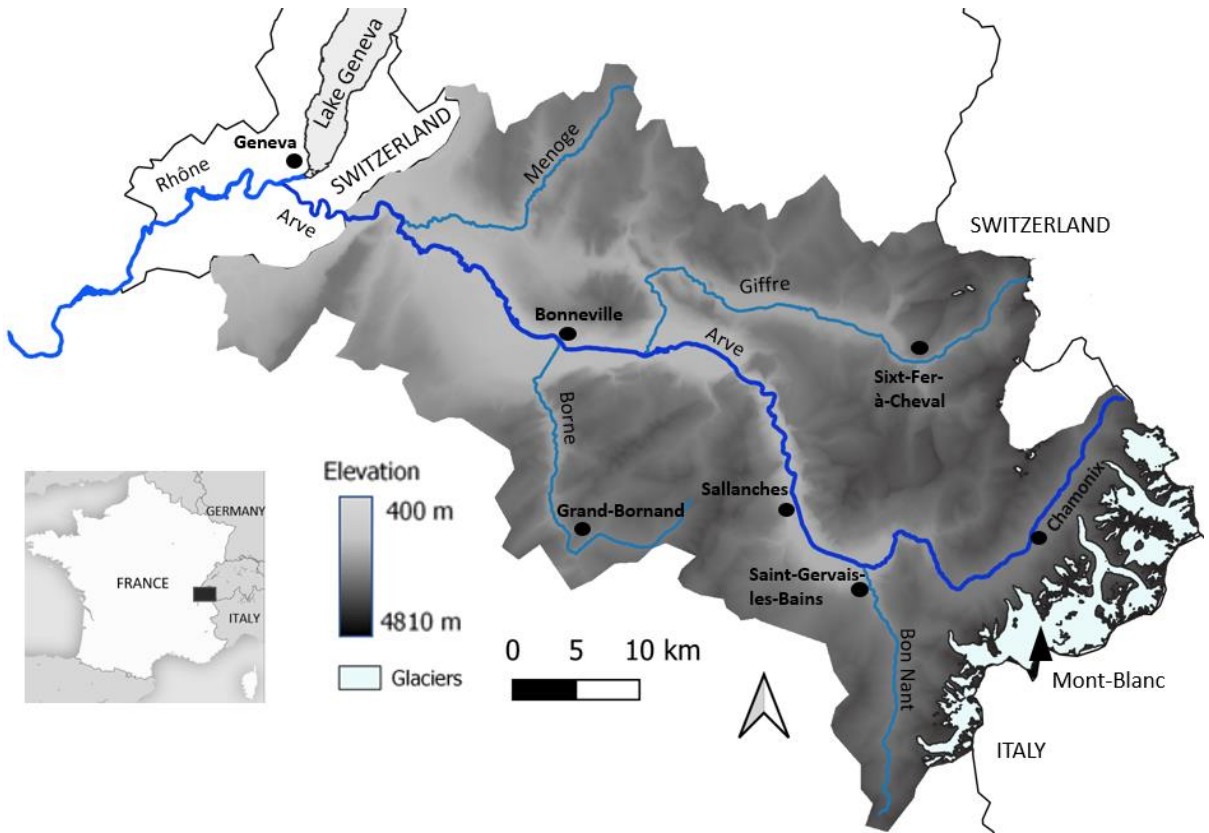

Figure 1. The Arve catchment location, topography, main hydrological network and the studied
cities (sources: IGN, 2017, 2015; SITG, 2020; GADM, 2018).
Due to large differences in altitude between high and lowlands, the Arve flows can be defined
by two hydrological regimes following an upstream to downstream continuum:
-  The upstream part of the catchment (down to the city of Sallanches; Figure 1), has a
glacio-nival regime due to the numerous glacial tributaries flowing from the Mont Blanc
massif (Viani et al., 2018). Low flows occur in winter and early spring (December to
March) and the high flows in summer (maximum in July and August) because of the
strong contribution of ice melt (Bernard, 1900). Floods mainly occur in summer due to
the synchronicity of both ice melt and intense subdaily rain storms. In this part of the
catchment, the flood plain is narrow and the slope inclination is high.
-  At lower elevations (i.e. downstream Sallanches) the regime becomes more nival
downward. Low waters mainly occur in winter and reach the highest levels between
late spring and early summer with the snowmelt. Between Sallanches and Bonneville,
floods mainly occur in summer and autumn due to the conjunction of intense rain storm,
snow melt and, in a lesser extent, ice melt contribution. Downstream Bonneville, floods
occur at any time of the year due to various hydro-meteorological interplays. The valley
floor is wide and may be affected by extended flooding.

**2.2.      Socioeconomic setting and land use.**
There are 106 municipalities located in the Arve catchment, with major population
growth since 1850. For instance, the population has been multiplied by a factor of three in
about 170 years at Chamonix (2 304 to 8 611 inhabitants between 1848 and 2016) and by a
factor of thirty-four in Annemasse (1 047 to 35 712 inhabitants between 1848 and 2017)
(INSEE, 2019). These numbers also hide large seasonal variations related to tourism activities.
This is particularly the case in Chamonix where the number of residents increases up to ten
times in high season. Most inhabitants live in the valley floor and foothills with most of the
farming, industrial and tourism activities as well as the main transportation routes and urban
areas. Since 1965 and the opening of the Mont-Blanc Tunnel Highway the Arve valley is a
major trans-Alpine route connecting France and Italy.
The socioeconomic setting of the valley follows an upstream-downstream distribution
pattern. The period from 1850 to 1913 experienced great tourist development (thermal bath of
Saint-Gervais-les-Bains and mountaineering in Chamonix). The economy around Chamonix is
essentially based on mountain tourism. In 1921, 250 000 tourists were visiting Chamonix each
year (Gex, 1924). In 2015 the lodging capacity in the valley reached about 416 400 equivalent
tourist beds. This part of the valley has undergone rapid urbanization. In 1804, the discovery
and exploitation of spring water for hydrotherapy in Saint-Gervais-les-Bains (Gex, 1924)
fostered the construction of tourist accommodations. Around Bonneville, the valley is a densely
populated corridor characterized by an old metal working industry, born from the watch
manufacture and nearby hydropower resources. The smaller valleys of the Arve tributaries are
sparsely populated and the economy is based on tourism and farming. Due to the
attractiveness of the city of Geneva, the valley from Bonneville to the Rhône River confluence
is characterized from the 1960's by a major population growth, extended industrial areas and
strong urbanization. Between the town of Cluses and Geneva the valley floor is almost
continuously built-up.

## 3. Material and methods of the HIFAVa database.

### 3.1.        Collecting data from historical archives.
Mélo et al. (2015) undertook historic research to develop a flood chronology and hydro-
meteorological circumstances of the flooding events which happened in the Arve catchment
between the 18$^{th}$ and the 21$^{th}$ century. As sources are more abundant and richer in information
over the last 165 years (1850-2015), this period was defined as the studied time frame of the
HIFAVa database. Only events that triggered impacts and that were mentioned in at least two
sources were integrated in the database. Since 2015, data have been further collected to
complete the preliminary dataset from Mélo et al. (2015).
Information on flood related impacts were collected from various sources. Primary sources
range from handwritten archives like municipal acts to departmental archives (e.g. reports of
the *Préfecture* and of town councils). Secondary and tertiary sources are respectively made of
published documents (newspapers, reports, books) and pre-existing databases. The database
of historical records providing a chronological and synthetic layout of the data is composed of
(Figure 2 and Table A1):
Manuscript materials:
-    Most of the manuscripts are kept in the departemental archives (*Archives*
*Départementales de Haute-Savoie*: ADHS) (Conseil départemental de la Haute-
Savoie, n.d.) or in the municipal archives (*Archives municipals* : AM) of the towns of
Chamonix, Cluses and Bonneville. The departmental and municipal archives also
contain records older than the Savoy annexation by France in 1860. The records can
be private or from a public institution.
Printed materials:

- Articles from scientific journals and books used in this study have been published since
1914. They mostly correspond to analyses of the regional hydrology (Mougin, 1914;
Rousset-Mestrallet, 1986) but also focus on single hydrological events (Pardé 1931;
Rougier 2002; Goy 2002) and risk assessment (Douvinet et al., 2011).
- Open-access online municipal risk prevention plans (*Plan de Prévention des Risques* :
PPR ; and *Plan de Prévention du Risque Inondation* : PPRI) (Préfecture de la Haute-
Savoie, n.d.) as preventive regulatory documents used to delineate risk areas, often
compile historical flood events that affected municipalities.

Newspapers:

- Most of the newspapers used are regional, but one is published at national level (*Le
Figaro*) and another is printed in Geneva, Switzerland (*Le Journal de Genève*). Most of
the newspapers can be found online or are kept in the departmental or municipal
archives. Newspapers describe the causes and consequences of the flood events and
sometimes provide instrumental data and illustrations. They also contain information
concerning the public authority response and past discussions.

Other records:

- The national database of historical flooding (*Base de Données Historiques sur les
Inondations* : BDHI) gathering floods events considered as "remarkable" in the French
territory (Ministère de la Transition Ecologique, n.d.; Boudou, 2015) was also used.
- The database created by the department of Restoration of Mountainous Areas
(*Restauration des Terrains en Montagne* : RTM) from the public institution managing
the French public forests (*Office National des Forêts* : ONF). The ONF-RTM database
gathers transcriptions of observations of the RTM officials as well as information
collected from diverse primary sources. These data (labelled RTM in the HIFAVa
database) are freely available through the open-access and online ONF-RTM database
(RTM and ONF, 2012). Some specific ONF-RTM reports are also included in the
HIFAVa. The RTM database was built to assist the management of small tributaries.
- A movie realized in 1990 by the RTM is also mentioned as a source.
- Some records are from the *Syndicat Mixte d'Aménagement de l'Arve et de ses
Affluents* (SM3A), which is the institution in charge of the management of the Arve River
and its tributaries since 1994.


### 258    3.2.      Characteristics of the HIFAVa database.

The database has been built using the free and open-source relational database
management system PostgreSQL and is accessible through its package pgAdmin.
HIFAVa contains 916 distinct flood impacts caused by 321 flood events. The primary key is
the impact ID. Therefore, each impact is recorded as a unique line and described through
various variables (Table 1 and Table A2). The river that triggered the flood is mentioned when
possible (94% of cases). For instance, no river name has been attributed to the impacts related
to overland flow in January 1979. The accuracy of the impact location varies from specific
addresses (house, bridge, neighborhood) to the municipality scale. When the source is not
accurate enough to distinguish distinct locations of several impacts, they are all referenced
under a unique impact ID. In other words, sometimes numerous impacts caused by a single
flood event are registered under distinct ID because it was possible to localize each impact
precisely (at the hamlet scale). Sometimes we can only localize the impacts at the municipality
scale, meaning that all impacts are registered under the same ID. The severity of an event can
not be estimated by the number of ID registered in the database. The most recent sources are
often highly informative, allowing impacts to be more precisely located.
Impacts occurring on the same day on a given river are expected to be caused by the
same flood event. As a result, the date is the key used to connect each impact to a flood event.

This "flood event" definition has been extrapolated to impacts occurring on the same day in different catchments, assuming that two impacts occurring the same day can be caused by the same hydrometeorological event given the moderate surface area of the Arve catchment. The accuracy of the date is rated on a certainty scale (hour, day, month, year). Based on information contained in the records, we distinguish when possible the hydrometeorological events (e.g. rainfall, intense and short rainfall, melting of snow, frozen soil, glacial outburst, wet period before the event) which caused the flood and the different flood types (e.g. river flooding, overland flow, sediment transport) leading to the impacts.

### 3.3.      Text mining.

The flood impacts of the HIFAVa were categorized through a text quantitative content analysis with the KH Coder software (Higuchi, 2015). The description of the impacts comes from comments contained in the records. The most frequent words were gathered in order to determine representative categories of the database content. A category is made of several words assigned to a coding rule. Categories were inspired by the flash flood impact severity scale of Diakakis (2020). This analysis led to the following seven categories with example of the assigned words:

- Transport network: e.g. "road", "bridge", "railway", "street".
- Urbanized areas and residential buildings: e.g. "house", "town", "basement".
- Natural environment: e.g. "forest", "field", "yield", "sediment transport".
- Protection infrastructures and dams: "dikes" and "dam".
- Industrial, commercial and recreational facilities: e.g. "mill", "factories", "golf", "camping", "hotel", "school".
- Critical installations: "drain", "power transformer".
- Victims: "dead", "injured", "missing", "evacuee".

| ID | Event | Sources | Start_day | Start_month | Start_year |
|---|---|---|---|---|---|
| 140 | 58 | Payot 1951 / Goy 2002 / RTM / BDHI | 12 | 7 | 1892 |
| 670 | 263 | RTM / Dauphiné libéré : 26/07/1996, 27/07/1996, 30/07/1996 and 02/08/1996 | 24 | 7 | 1996 |

| Start_date | Time_unc | Hour | River | Impacts | Municipality |
|---|---|---|---|---|---|
| 12/07/1892 | D | | Bon Nant | Thermal Bath, victims | Saint-Gervais-les-Bains |
| 24/07/1996 | H | 11 pm | Arve | City center | Chamonix-Mont-Blanc |

| Impact_latitude | Impact_longitude | Space_unc | Count | Hydrometeo_descript | Water_level |
|---|---|---|---|---|---|
| 45.8965 | 6.70596 | A | 175 | No entry | |
| 45.9257 | 6.87057 | A | 0 | No entry | 0,4m |

| Flood_description | Impacts_decription |
|---|---|
| Debris flow. 30 minutes to flow from the Tête Rousse glacier to the Arve confluence. | More than 175 victims, destruction of most of Thermal Bath, mud over the first floor. |
| Debris flow, logjam under bridges. | Numerous houses and the city center are under water. Fuel oil pollution due to basement flooding. |

Table 1. Extract of the HIFAVa database showing its structure and examples of content. Refer to the Online Resource 2 for details about the column contents.

## 4. Results and discussion.

In this section, we present the detailed content of the database, the results of the preliminary analyses and the first key findings.

311

### 4.1.   Evolving sources over time.

During the studied period, the diversity and the quantity of sources in which mentions of impacts were found fluctuate (Figure 2). Among the existing databases used, the BDHI database for instance continuously covers the studied period but was sporadically informative since it only contains two mentions of impact events in the Arve catchment, respectively of the Bon Nant River in 1892 and of the Borne River in 1987 (Figure 2.a). By contrast, the SM3A database appears later (1979) in the studied time frame.

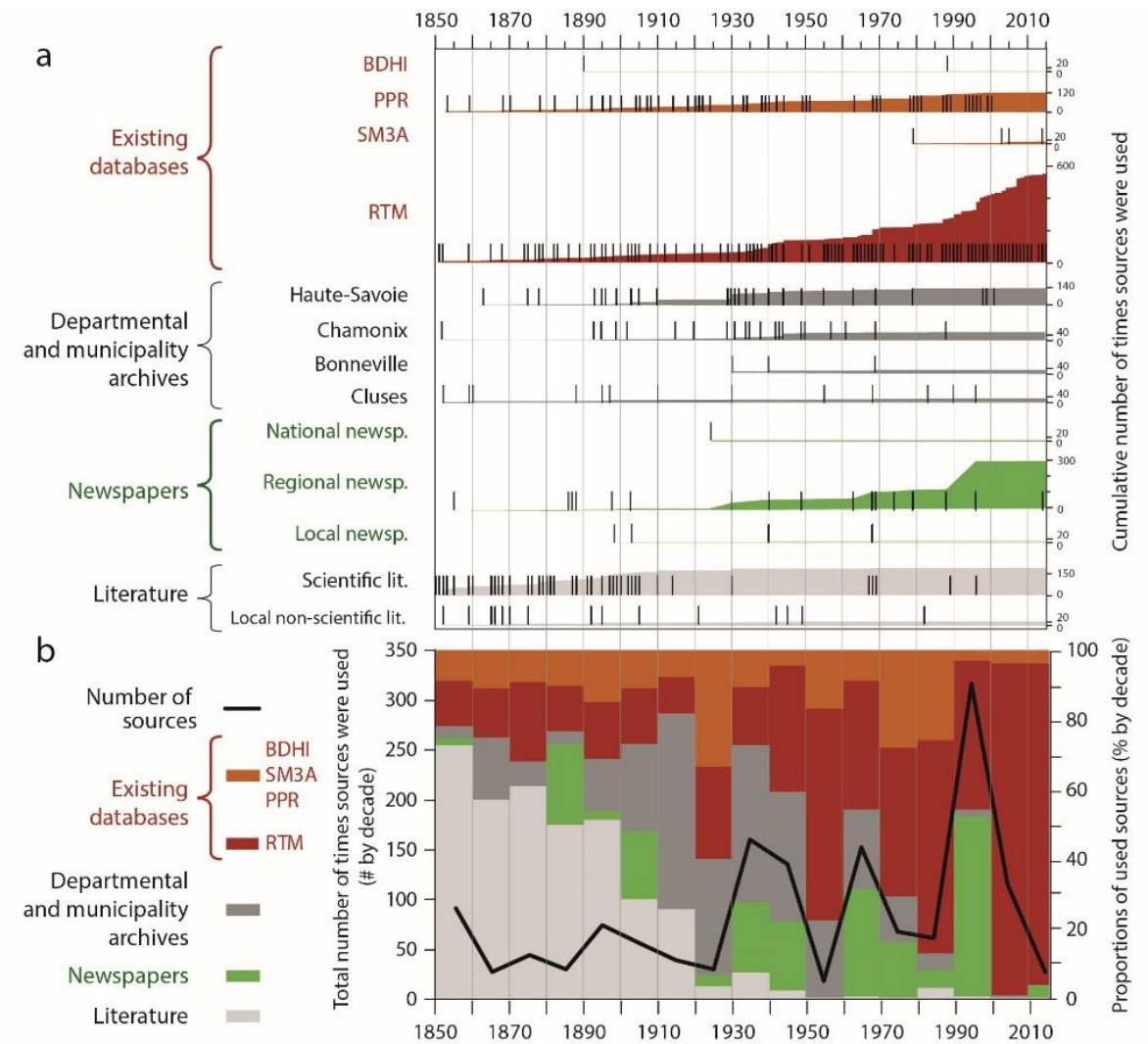

Figure 2. a) Number and b) percentage of mentions from the studied sources to document flood impacts since 1850 in the Arve Valley. In panel a, the ticks indicate each mentions of impacts and the colored areas display the cumulative number of mentions.

RTM (38% of the mentions from all sources), regional newspapers (20%) and scientific literature (13%) constitute the main sources of information on past flood impacts throughout the studied time period. Sources from the literature are particularly useful (54%) to document the period between 1850 and 1910. Then, materials from the departmental and municipality archives are abundant between 1900 and 1970, especially those from Haute-Savoie (20% of the total registered sources for this period) and Chamonix (7%). Between 1940 to 2015, the RTM represents 58% of the mentions from all sources describing the impacts (Figure 2.b).

One of the evolutions of the sources is the increase in newspaper articles mentioning flood impacts. Following the 1881 press freedom French law, the 1880-1889 decade marks the emergence of articles recording natural hazards, such as flood impacts (Ferenczi, 1996). However, the 1855 flood in Bonneville was already reported by the swiss newspaper, *Le Journal de Genève*.

Although few sources (e.g. the municipal archives of Sallanches) remain to be examined, we consider that most of the main sources (newspapers, existing databases, and public archives) have been analyzed. As a result, we are confident that no event that was deemed damageable by local communities was missed and we consider that we have a comprehensive view of past flood impacts since 1850 over the whole Arve catchment.

### 4.2.    Changes in impacts over time and space.

From 1850 to 1920, the number of impacts fluctuates and only four years are considered remarkable with more than 15 impacts (1852, 1878, 1895 and 1910). From 1920, years with 15 or more impacts become more frequent (1930, 1940, 1944, 1968, 1979, 1987, 1990, 1996, 1997 and 2007) and the total amount of impacts per year reaches the maximum value of 54 in 1996 (Figure 3). The decennial moving average of the impacts' number highlights an overall increase over the 165 years, punctuated by periods with less frequent impacts (in 1910-1923, 1950-1960 and 1975-1980).

The number of recorded impact events – i.e. flood events recorded in the historical sources because of the impacts they caused – stays relatively stable between 1.5 and 3 events per year on average until 1990, then it rises up to 4.5 events per year. As hydrological data are available from 1950 for the Arve River, it is not possible to assess whether the increase in impact event was a function of increased flood occurrence or changes in vulnerability or recording.

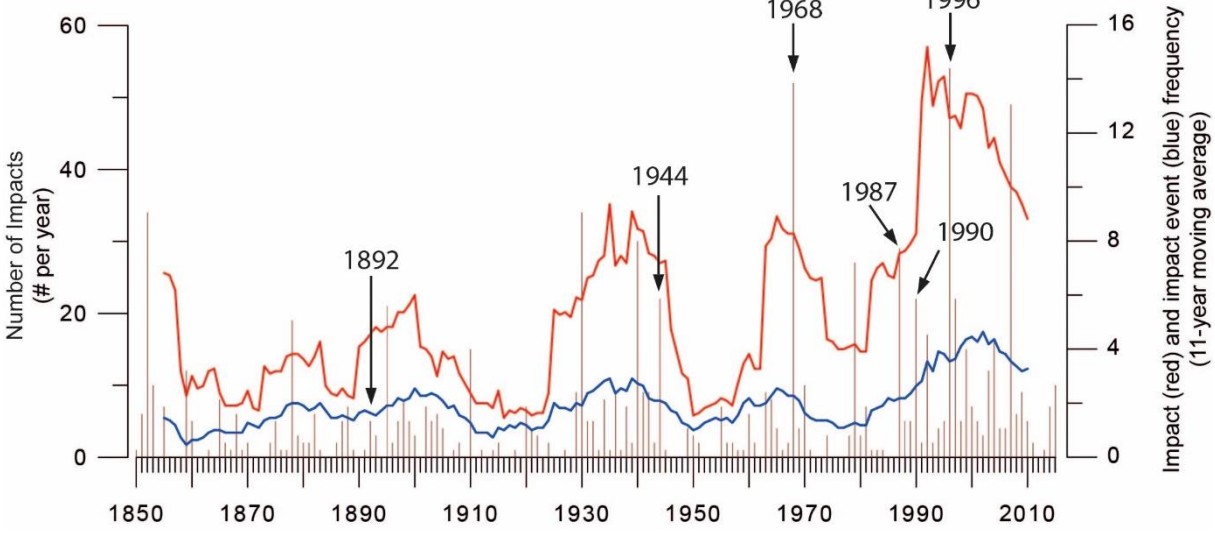

Figure 3. Yearly occurrence of impacts and decennial moving averages of impacts (red curve) and associated impact events (blue curve). Events discussed in the article are labeled with their year of occurrence.

When analyzing the spatial distribution of the flood impacts, we can see that they are spread over the entire catchment (Figure 4a). Chamonix and Bonneville gather, however, respectively 24% and 12,5% of total impacts recorded in the Arve catchment. These high numbers may be due to the fact that these towns are both among the most densely populated and the closest to the Arve River. The impacts caused by the Arve River floods represent 33% of all recorded impacts, and its  two main tributaries, the Giffre and the Borne Rivers, have

only caused 8% of the recorded impacts (Figure 4b). In fact, most impacts are due to small torrential streams (53%). Among them, almost a third are related to glacial tributaries, while these tributaries are localized only in the uppermost part of the catchment near Chamonix. For instance, small torrential tributaries such as the Arveyron, the Grépon (left bank tributary close to Chamonix) or the Bon Nant caused alone more impacts than the Borne River itself.

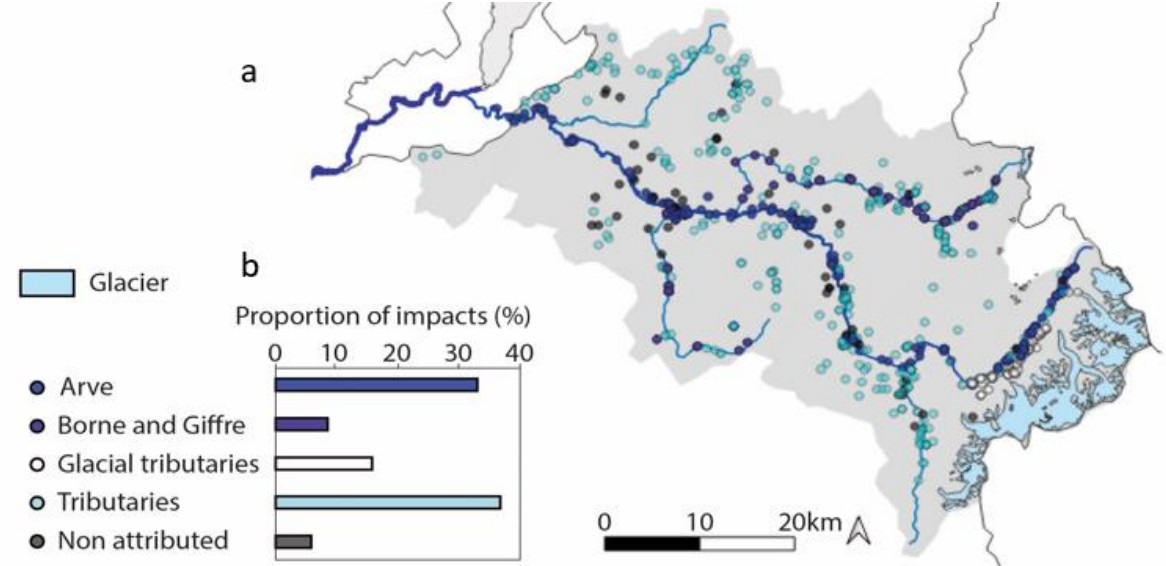

Figure 4. a) Location and b) distribution of flood impacts caused by the Arve River and its tributaries. The category "non-attributed" corresponds to the impacts for which it was not possible to attribute a river, either because events are related to overland flows or because the source did not mention the river (sources: IGN, 2017, 2015; SITG, 2020; GADM, 2018; HIFAVa).

The Arve tributaries produced disasters characterized by numerous and major flood damage. Among them, the 1987 Borne River flooding in its uppermost part washed away the municipal campsite of the village of the Grand-Bornand causing 23 casualties and heavy economic losses (Meunier, 1990). In addition, the 1892 glacial lake outburst from the Tête Rousse glacier in the Bon Nant River (which literally translated means "Good Stream") swept away the thermal bath of Saint-Gervais-les-Bains (Figure A1) and 33 houses causing at least 175 casualties. The glacier was drained in 2010 and is today closely monitored to avoid such a brutal and disastrous natural event (Garambois et al., 2016; Vincent et al., 2012).
All these high impacts events are due to sudden, highly-dynamic summer floods of tributaries, often aggravated by large sediment transport. Some towns located along the Arve River – such as Sallanches – are more prone to tributaries floods because embankments have been built and efficiently prevent impacts from the Arve. In contrast, there are very few impacts recorded in high altitude locations, probably due to the sparsity of human settlements.

### 4.3. Potential drivers of changes in the number and distribution of impacts.

The increase in the number of impacts starting in the 1920's and well-marked from the 1960's can be explained by multiple factors, such as an increase in flood occurrence and/or magnitude, a source effect, an increase in exposure of goods and people, a deterioration of the diking systems, a break in the risk memory transmission, an evolution of the risk perception or an evolution of the local political risk management. Due to the lack of available data regarding changes in flood hazard, protection infrastructures and the risk memory, perception

and management, we only explore whether source effect and changes in exposure may explain the observed increase in impacts.

To decipher the potential source effect in the increase in impacts particularly noticeable since 1960, maps of the impacts by sources have been drawn for the periods before and after 1960 (Figure 5). In addition to the noticeable increase in impacts, this date marks a strong change in the Arve valley economy, from a rather homogeneous agricultural society to an industrial society exploiting the river bed for materials extractions. After 1960, the Arve watershed also experienced a strong tourist development and a rapid demographic expansion. From 1850 to 1959, three main sources describe 64% of the impacts (literary records 28%, RTM 18% and departmental and municipal archives 18%) and for 29% the information comes from more than one source. The impacts are mainly gathered along the Arve and the Giffre Rivers, especially in the valley of Chamonix and between the towns of Cluses and Bonneville. For the second period (1960-2015), the RTM reports 65% of the impacts, and 20% come from multiple sources while departmental and municipal archives and the PPR/PPRI describe 5% each. Information coming only from literature decreases substantially (122 described impacts in the first period to 3 impacts in the second), SM3A records start in 1979 and only document the Giffre and the Bon Nant Rivers. The distribution of the impacts is much more scattered across the whole catchment than during the first period (Figure 5). The impacts are not gathered along the Arve River, since most of them result from small tributaries. Impacts described by more than one source are located in the valley of Chamonix and around Bonneville, probably because these economic and tourist centers arouse interest of many sources (newspapers, departmental and municipal archives and RTM). In addition, the strong emergence of the RTM since 1940 (Figure 2) can explain the rise in documented impacts caused by small tributaries (Figure 5). Following the floods of the Rhône, the Loire and the Garonne in 1854, the 1858 law against urban flooding places flood control at the heart of the national legislation for the first time. In the following, the RTM department was formed for the reforestation of mountains slope in order to prevent the reproduction of major floods. The department became quickly efficient and since 1860 collected numerous reports. Built for the study and management of small tributaries, the RTM database became the main source of information since 1930 for the HIFAVa database. Hence, the strong emergence of the RTM source among the others may play a role in the observed increase in impacts since 1920, even more noticeable since 1960.

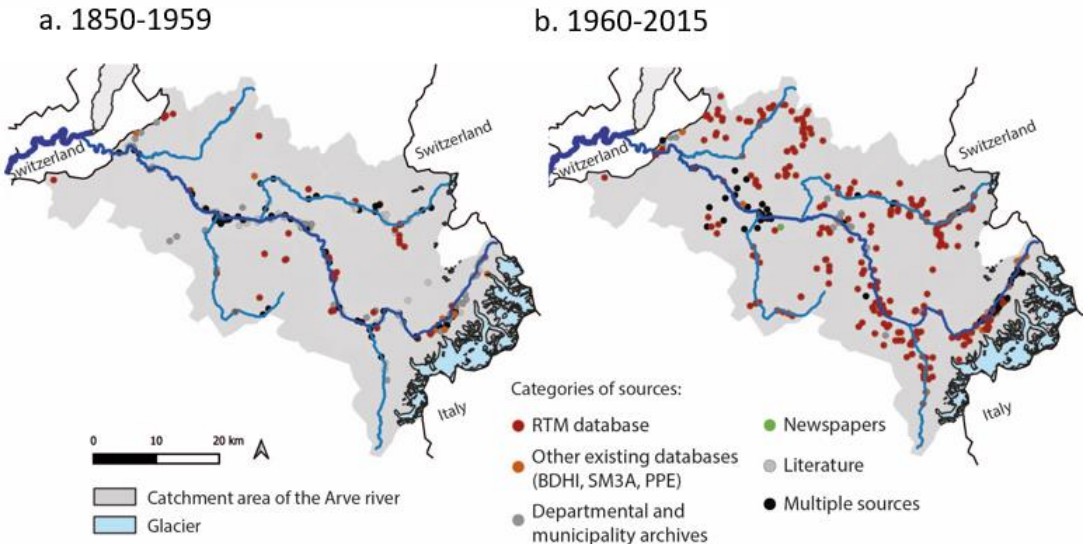

Figure 5. Comparison of the spatial distribution of impacts by categories of sources in the Ave catchment during the periods (a) 1850-1959 and (b) 1960-2015 (sources: IGN, 2017, 2015; SITG, 2020; GADM, 2018; HIFAVa).


442    The rise in the number of impacts per flood may be partly explained by the fact that
distinct impact types in the same location were reported and therefore referenced under
distinct impact ID, while they were not differentiated in previous periods. Recent sources seem
to provide more accurate information on the impacts and their locations. In older sources,
impacts are most of the time documented at the city scale (21% of the impacts for the first
period, and 10% for the second period). Thereby, all these impacts are stored in the database
in a single line with an uncertainty code for the impact location corresponding to the municipal
level. In most recent sources, impacts' locations are described more accurately allowing them
to record at a resolution up to the building scale. As a result, impacts are stored in as many
lines as impacts locations can be identified, with an uncertainty code for the impact's location
corresponding to the building or neighborhood level (85% of the impacts for the second period).
For example, in 1996 fifty-three impacts where recorded for the same event and fifty of them
where located in distinct places in Chamonix. The rise in impacts can also be due to numerous
impacts in different locations, as the flood of 1990 which impacted six towns in two different
sub-catchments (the Arve and the Giffre catchments). However, in order to overcome the bias
induced by the recording of impacts according to their location, we aggregated the impacts at
the municipality level. That is to say, all the impacts reported for a given municipality that were
caused by the same event (thus the same day) are recorded under the same line in the
database. This results in 562 "aggregated" impacts instead of 916 impacts initially. From these
data, we redrawn Figure 3 (Figure A2) comparing the moving average of impacts and
associated events. We can see that in both figures the trends of increasing impacts are similar.
There is an increase of impacts (here starting soon as 1890s). Thus, the way the impacts are
stored in the database (by location or by municipality) affect the absolute values of impact per
year but not the observed temporal changes over time.

467    Changes in exposure and vulnerability related to land use is another potential
explanation of the rise in the number of impacts (Magnan et al., 2012; Garnier and Desarthe,
2013; Camuffo et al., 2020). In fact, as major population growth happened, especially in
Bonneville and Chamonix, and lead to a strong and fast urban sprawl in the flood plain between
the 1950's and the 2010's as shown by aerial photographs (Figure 6). They also show the
vanishing of the alluvial forest (Dufour and Piégay, 2006) and cultivated fields to the benefit of
urbanization in both towns. Upstream, in Chamonix, the demographic expansion dates back
to the early 20th century with the flourish of mountain tourism. In downstream towns – e.g.
Bonneville and Annemasse – the expansion starts in the 1950's because of the economic
attractiveness of Geneva.

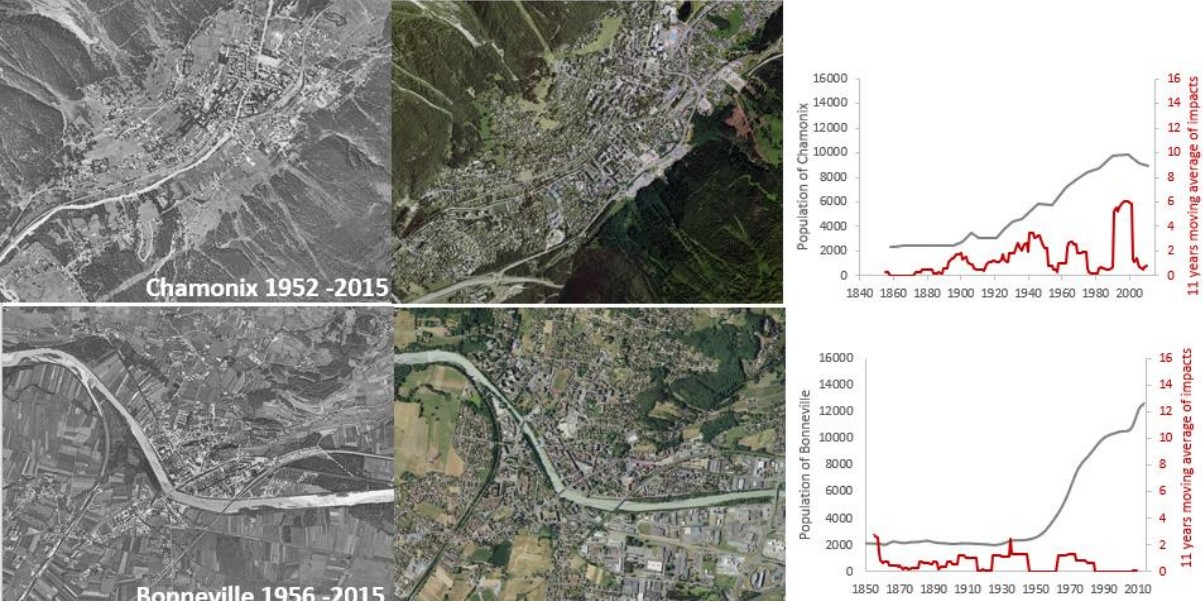

Figure 6. Aerial photographs highlighting changes in land use and urban sprawl growth in
Chamonix and Bonneville as well as plots stressing changes in impacts and population growth
from 1848 to 2011 (IGN, INSEE, HIFAVa).

Besides these numbers, the urban expansion and the growth in tourism entail the arrival of
new residents in the valley (INSEE, 2020), unaware of the local hazard history. The valley
narrowness, the demand for land and the loss of memory of past events have led to a rise of
constructions within historical flood-prone areas, resulting in an increased exposure. For
instance, in 1944 recently built houses in Chamonix were washed away by the Grépon River.
The same situation also happened during the 1968 flood that destroyed a new residential area
in Bonneville. To explore to what extent the urban growth may have resulted in an increased
exposure explaining the observed rise of impacts, we use the population growth data as a first-
order proxy of urban growth. In Chamonix, the number of impacts increase from year to year,
somehow mirroring the population growth (Figure 6). Therefore, population exposure might be
one explanation for the increased number of impacts. One can, however, notice the decrease
in impacts in the early 2000s due to the heightening of the dikes after the 1996 flood. In
Bonneville, the link between the number of impacts and the population growth is not as clear
as in Chamonix. This may partly result from the absence of major floods since 1968. Therefore,
an increasing exposure might locally explain the increase in impacts. Further studies should
however reconstruct diachronic maps of land use to assess in a finer way the urban growth in
flood-prone areas and its link with changes in impacts number. Overall, the potential role of an
increased exposure is not excluding the indirect sources affect (emergence and dominance of
the RTM source), but both factors can be combined and complement each other.

**4.4. Nature of the flood impacts.**
The quantitative analysis of text content reveals the distribution of the impact categories
by river and illustrates the diversity of the catchment in terms of land use and economic
development (Figure 7). This analysis of text content is particularly relevant because it allows
to overcome the database scarcity of quantitative information on the severity of the flood.
Indeed, it is difficult to estimate the severity of a flood event as the flow rate and water height
are only mentioned in rare cases. However, according to Barriendos et al. (2019) a link can be
made between the nature of the impacts and the severity of the triggering phenomena. In order
to establish this link, we also need to consider the various dynamics of the flooding phenomena
as fast flooding generally affect smaller surfaces but in a more violent manner than slow floods

of the main rivers. Therefore, they may trigger a lesser number of impacts but the level of destruction of the impacted element might be more important. Of course, the level of destruction also depends on the sensitivity and physical vulnerability of the exposed element, it is why categorizing impacts by their nature might help exploring the question of the flood severity.

The categories used for the analysis are partly inspired from a recent paper from Diakakis (2020) and refined based on textual analysis of the words used to describe the impacts in the database. The number of occurrences of the words determining the category is here named as the number of mentions. For the overall Arve catchment, damage to the transport network is the most frequent (406 mentions), followed by damage to urbanized areas (286) and natural environment (253). Impacts on transport network are in proportion equally distributed among river types (all around 30%). Impacts on industrial facilities are mainly caused by the Arve and the Giffre Rivers as their wider valleys allow the installation of these facilities. The Giffre and the Borne Rivers have caused the least impacts specifically on urban areas, probably because of less dense population and of an economy based on farming and tourism activities.

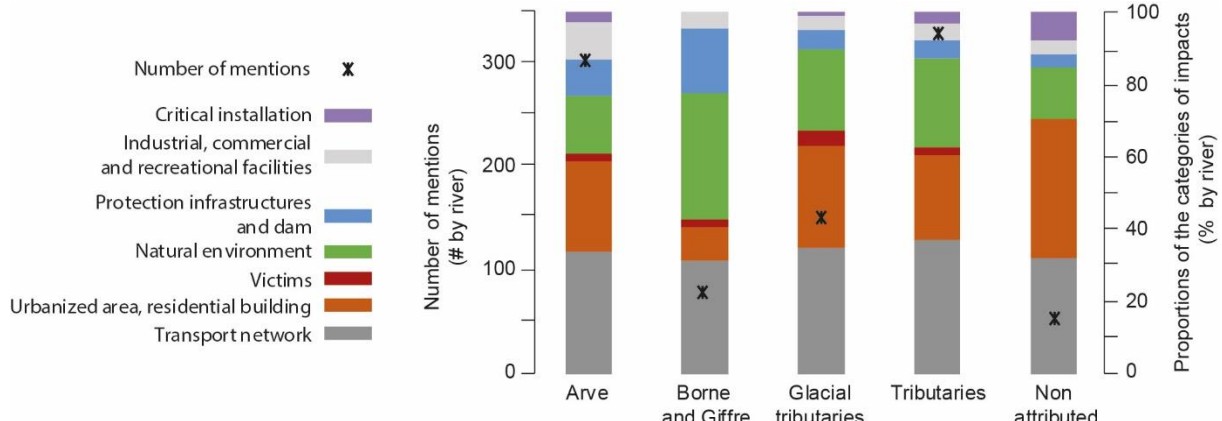

Figure 7. Distribution of flood impacts categories according to the river types. The class "non attributed" gathers the impact that could not be assigned to a river (e.g. overland flows).

Most of the mentions of victims refer to impacts caused by tributaries (20 out of 28) characterized by faster hydrological responses. They have caused 80% of the deadly impacts registered since 1850 in the whole catchment, e.g. the 1892 glacial lake outburst and the 1987 flood of the Borne River in Grand-Bornand. The mentions of victims of the Borne River should belong to the small tributaries class as the impacts occurred in the uppermost part of the catchment. As shown by Jonkman et al. (2005) and Ruin et al. (2008) high mortality rates are mostly due to the suddenness and violent responses of small catchments affecting people in the open air (as for campers).

To track potential changes in the nature of impacts since 1850, they are scrutinized by decades over the last 165 years (Figure 8). Impacts on transportation networks are present in every decade since 1850 but they increase after 1930 as well as impacts on urbanized area and natural environment categories. It was not until 1960 that impacts on the industrial, commercial and recreational facilities category increased. Mentions about critical installation (sewers and water pipes) are recorded for the first time in the 1960's. Mentions of victims are present in almost every decade.

Nevertheless, no major evolution of the impacts' categories can be seen, except the emergence of mentions of critical installations in the 1930's. A more in-depth analysis will be

conducted later on to define severity classes based on the nature of the impacts and also to
identify whether there is an evolution of the lexicon used to describe the impacts.

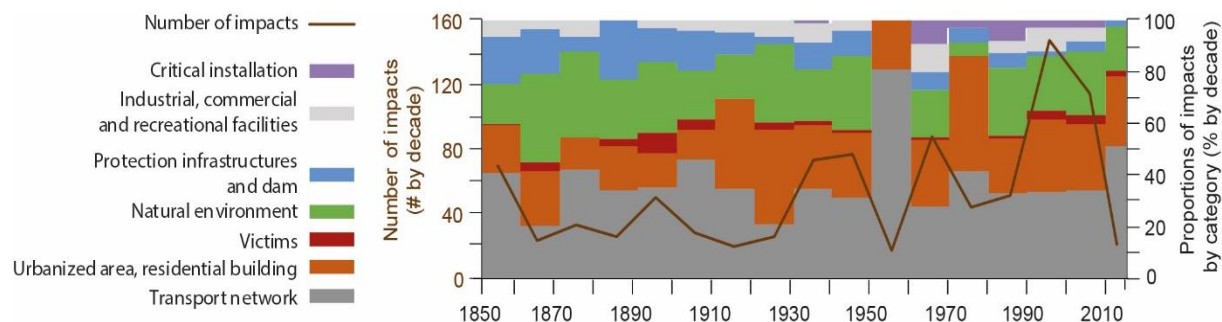

Figure 8. Decennial histogram of the evolution of the categories of impacts and the number of
events.
The data representation of Figure 8 does not allow to visualize the evolution of the absolute
values. For instance, at the catchment scale there is an increase in the number of mentions of
the victim category with 18 mentions during the period 1960-2015 compared to 10 for the
period 1850-1959 (Figure 8). This increase is not easily noticeable when looking at proportions
of impact categories because of the strong augmentation in the total number of mentions since
1930. Yet, apart from the disaster in the Grand-Bornand in 1987 (Borne River), since 1931 all
the mentions of victims refer to evacuations, recues or injuries.
The number of impacts has been almost multiplied by two since 1920. Mentions of impacts on
urbanized areas during 1960-2015 has been multiplied almost by four compared to the 1850-
1930 period. This agrees with the evolution of the land use due to the demographic growth,
i.e. the observed vanishing of forest areas and cultivated land to the benefit of urbanization
(Figure 6). The mentions of impacts to natural environment for the period 1850-1930 are more
than double compared to the period 1960-2015. During the first period, mentions of impacts to
natural environment refer mainly to forest, field and crops, while after 1960 there is no mention
of field or crops and most of the mentions are about gullying, deposition of sediments and
banks.

## 5. Conclusions.
This paper describes and analyses historical data documenting 916 flood impacts
associated to 327 flood events that occurred between 1850 and 2015 in the Arve valley, an
Alpine catchment characterized by a high hydrological and socioeconomic diversity. This
dataset is stored in the HIFAVa database fully described in this paper.
A first spatial and temporal analysis of this dataset allowed to identify three key findings
and research perspectives:
- The predominance of impacts due to torrential tributaries. There are two main types of
flood events causing impacts in the Arve catchment, e.g. floods related to the main
river and those related to the smaller mountain streams. Floods from these small
streams are characterized by sudden and fast hydrological responses and most of the
time by high volumes of sediment transport, making hazard management difficult. They
caused two third of the 916 recorded impacts with numerous casualties, such as the
Bon Nant (1892) and the Borne (1987) Rivers floods. In contrast, slow rising, day-long
floods of the Arve River affect larger areas and trigger large economic consequences

but no casualties. The 1968 flood event affecting a large part of the Arve catchment is an exemplary case of this flood type. As suggested by Ruin et al. (2008), the number of impacts caused by torrential streams being much higher than those triggered by the main rivers calls for a greater attention to flood risk assessment and management in small catchments.

- The rise in the number of impacts starting in 1920 and well-marked from 1960. This increase in impacts may be explained by various factors. Based on the available data, we discussed the potential source effect and changes in exposure. It appeared that the emergence and dominance of the RTM among the other sources as well as an increased exposure linked to urban expansion may play a role at some places. However, exposure was assessed through the population growth as a first-order proxy of urban expansion. A more detailed study of changes in land use based on e.g. old maps and aerial photography is necessary to confirm this preliminary result.

- The evolution of the impacts' nature (increase of impacts on urbanized area) mirroring the land use changes and probably due to the urban expansion linked to the large demographic growth in the catchment area. Further work is required to explore the evolution of the vocabulary used to describe those impacts across centuries in order to evaluate how this evolution might relate to changes in what the societies values and care for across history. Another path of future research concerns the identification of severity classes based on the description of the nature and extent of the damage in order to be able to characterize the level of impact on a given territory, to allow a classification of past events according to their intensity and to identify the most significant ones.

Moreover, the other drivers that may have induce the observe increase in impacts still need to be investigated. The lack of gauge data (available only for the main river and since 1950) precludes the study of potential links between the increased number of impacts and changes in flood occurrence and/or magnitude over the whole catchment. In contrast, data on the risk memory and its transmission within the society can be acquired and analyzed to explore the evolution of the territory's vulnerability through time.

**Appendixes**

Table A1. List of the historical records – or their origin – used to provide information about flood events and impacts collected in the HIFAVa database. The date mentions the year of publication for books and period covered by the newspapers.

| Manuscript materials | Printed materials | Newspapers | Other records |
|---|---|---|---|
| ADHS | AM Cluses | ADHS | BDHI |
| AM Bonneville | Conard 1931 | Le Messager (1965-1968) | RTM (movie 1990) |
| AM Chamonix | Douvinet 2011 | Le Dauphiné libéré (1963-2014) | SM3A |
| AM Cluses | Goy 2002 | Journal de Genève (1855-1978) | |
| | Mestrallet 1986 | La Croix 74 (1898) | |
| | Mougin 1914 | L'Allobroge (1903-1940) | |
| | Parde 1931 | Le Faucigny (1968) | |
| | Payot 1951 | Le Figaro (1924) | |
| | PPR | Le Messager (1940) | |
| | PPRI | Le Mont-Blanc républicain (1903) | |
| | Rannaud 1916 | Le Progrès (1898) | |
| | Rougier | L'Industriel Savoisien (1910) | |
| | RTM | La Revue Savoisienne (1887-1889) | |

Table A2. Presentation of the HIFAVa database showing its structure with each data entry.

| id | [PK] serial | Primary key. ID number of an impact. |
|---|---|---|
| event | integer | Number of the event that triggers the impact. In case of an hydro-meteorological event, several impacts located at different places on different river are associated to the same event. |
| sources | integer | The different sources that provide information. |
| start_day | integer | Start day of event. |
| start_mont | integer | Start month of event. |
| start_year | integer | Start year of event. |
| start_date | date | Start date of event. |
| hour | text | Start hour of event. |
| time_unc | text | Uncertainty of the start date. H means that the start date is known at the hour scale, D at the day scale, M at the month scale and Y at the year scale. By default, when the day and/or the month is not known, "1" is attributed to start_day and/or start_month. |
| duration | real | Duration of event |
| river | text | River that trigger the flood (the cell may be empty if the impacts are not related to river flooding). |
| impact | text | Nature of impact. |
| municipality | text | Municipality where the impact is located. |
| imp_lat | real | Latitude of the impact in decimal degrees. |

| imp_long | real | Longitude of the impact in decimal degrees. |
|---|---|---|
| Space_unc | | Describe the spatial scale of impact location. A means that the impact is located at the scale of a point on a map, B at the scale of a part of a city, C at the scale of a city, D at a coarser scale than the one of a city. |
| count | integer | Number of victims |
| hydrometeo_descript | text | Description of the hydrometeorological event according to the sources. |
| precipitation | real | Precipitation rate given in the sources (mm). |
| flood_descrpt | text | Description of the flood from the sources |
| river_water_level | real | Water level of the river (m). |
| water_level | real | Water level of flooded area (m). |
| discharge | real | Discharge of the river (m3/s) |
| impact_descrpt | text | Description of the impacts according to the sources. |



Figure A1. The wiped-out thermal bath of Saint-Gervais-les-Bains after the debris flow of the 12th of
July 1892 (Thermal bath establishment of Saint-Gervais, via www.thermes-saint-gervais.com).

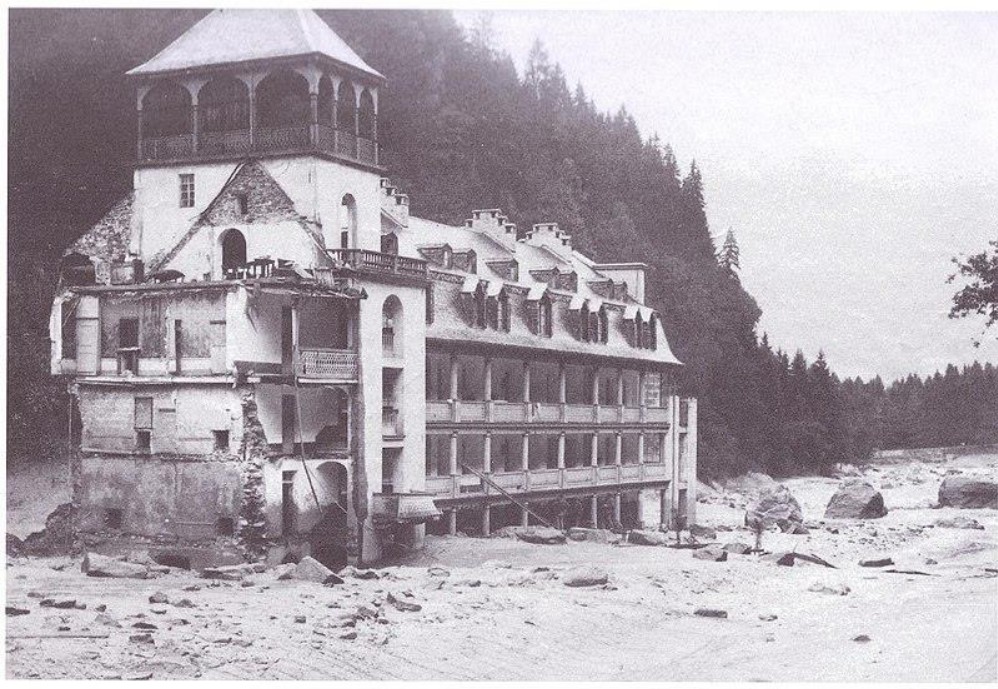



Figure A2. Yearly occurrence of impacted municipalities and decennial moving averages of impacted
municipalities (orange curve) and associated impact events.

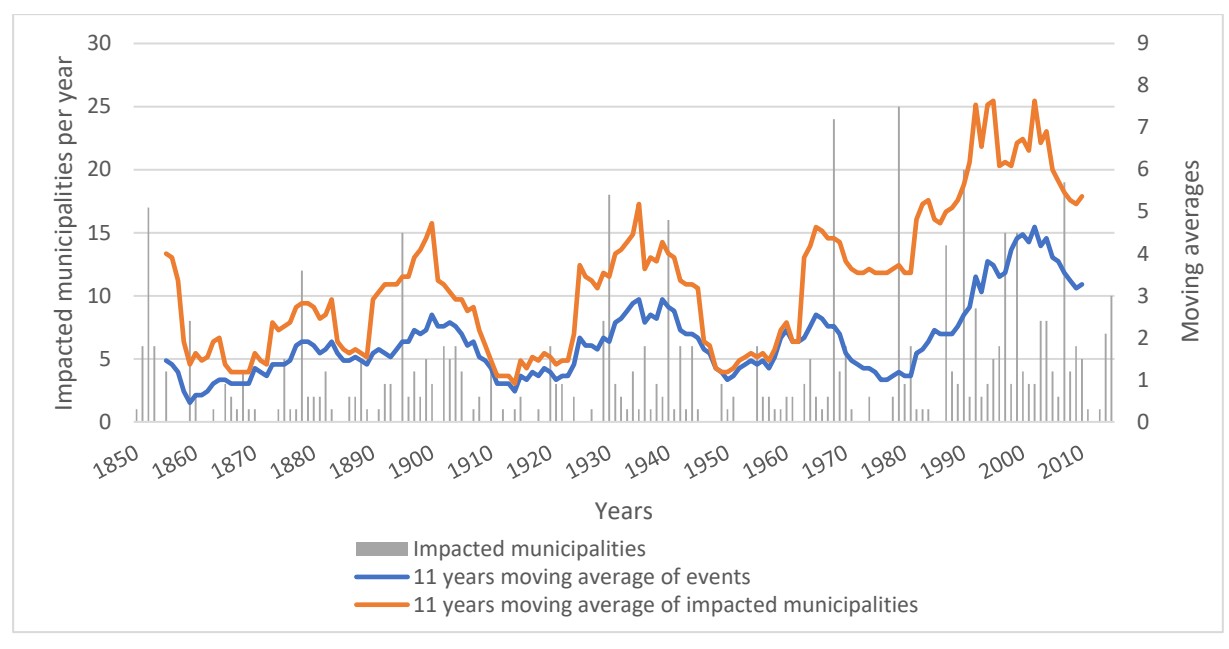

Acknowledgments: We would like to thank the *Syndicat Mixte d'Aménagement de l'Arve et de*
*ses Affluents* who shared its database of events and impacts. The HIFAVa database was partly
fed thanks to the funding of the projects Labex ITEM CrHistAl and OSUG2020 RHAAP. This
study was developed in the framework of *CDP-Trajectories* of the Grenoble-Alpes University
(France) funded by the *Agence Nationale de la Recherche* through the project
"*Investissements d'avenir*" (ANR-15-IDEX-02).

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
