# Peer review of "Geo-historical database of flood impacts in Alpine catchments (HIFAVa database, Arve River, France, 1850 – 2015)."

_Natural Hazards and Earth System Sciences, 2021_

## Referee Comment (RC2)

[referee-annotated manuscript omitted]

---

## Author Comment (AC1)

Dear Referee,

Thank you for your thoughtful comments, the time and effort towards improving our manuscript. We appreciate your comments on the figures.

We incorporated changes to reflect most of the suggestions you provided. Here is a point-by-point response to your comments and concerns.

As you wondered in your introductive commentary, this manuscript aims indeed to serves as a reference for describing the HIFAVa data set and its collection, further analyses are in progress. We completed the introduction to make this clearer.

In addition to the below comments, all spelling and grammatical errors pointed out have been corrected.

**Abstract.**

- **Comment 1:** *I suggest to streamline the beginning of the abstract, which is a little bit clumsy. Some examples:*
*l. 24: suggest to specify "it"*
*l. 25: the "Alps that warm at a rate twice as high in the Northern Hemisphere" - although this will probably be understood correctly by informed readers, the statement itself is somewhat unclear*

As suggested, we streamlined the abstract in order to make it more understandable. l.24 we specified "it" and we cleared the statement l.25. : *"In France, flooding is the most common and damaging natural hazard. Due to global warming floods are expected to globally exacerbate, and could be even more pronounced in the European Alps as the temperatures have been rising at a rate almost twice as high as the average in the Northern Hemisphere since the beginning of the 20th century. To approach long-term evolutions of past flood occurrence and related socio-economic impacts in relation to changes in the flood risk components (i.e. hazard, exposure and vulnerability), the study of historical records is highly relevant, especially in the context of the a densely populated area gathering at stake human and economic activities."*

**Introduction.**

- **Comment 2:** *l.118: I suggest to be more precise as far as the "first database documenting a mountainous catchment" is concerned. There are definitely databases on historic flooding and debris flows in other countries in the European Alps, including Switzerland, Austria and Italy. These do comprise natural hazard events on a catchment level, and some have been linked to exposure and mitigation (e.g. https://doi.org/10.1016/j.crm.2021.100294 for torrential flooding; https://doi.org/10.1016/j.gloenvcha.2020.102149 for avalanches), therefore effectively covering the "interactions between social and natural dynamics engendering flood impacts".*

Thank you for your reading recommendations that are very relevant to our study. To reply to your comment, we would like to point out that the specificity of this database is that it was developed from the historical impacts identified during the research in the archives, it is a collection of societal markers of the effects of floods. Most databases are built from hydrological data. However, we reworded this sentence: *"The study of this database ultimately aims at analyzing the interactions between social and natural dynamics engendering flood impacts. In this paper describes the HIFAVa data set and its collection."*

- **Comment 3:** *The term "impact" is a core concept throughout the manuscript. I think a clear definition on what an "impact" actually comprises would be helpful (either in the introduction or in section 3). This is somewhat hidden in section 3.3, I suggest to state this more clearly earlier on.*

We agree. By "impact" we mean any disturbance caused by a flood that has been reported in the varied historical sources available over the extend of the studied period. It can be the flooding of a field, the cutting of a communication route (footbridge, road etc.), the destruction of a building or victims.
An impact entry does not always represent an isolated damage, but can sometimes group together several imprecisely described and uncountable damages of different nature. When the source is not accurate enough to distinguish distinct locations of several impacts, they are all referenced under a unique entry.
We have added a definition in the introduction: L.70. "*The impacts recorded in the database can include any types of human goods that has been damaged by the inundation, like a field, a communication route or a building for example.*"

**Study area.**

- **Comment 4:** *Figure 1: Please rework the elevation legend. This is a continuous scale and should be presented as such. If a discrete scale is used, intervals need to be reported and not scalar values. Which color does an elevation of say 1000 m a.s.l. correspond to? Is it green or yellow, or something in between? In addition, I would advise to use prettier breaks, and not 401 - 1401 - 2902 - 4810, which seems somewhat arbitrary. The color of the Arve river in the legend does not correspond to the color in the map.*

We agree with this comment and modified the figure accordingly by using a continuous scale in grey shades to ensure the legibility of the other layers of information. We also changed the color of the Arve river to make it more discernible.

[Figure]

Figure 1. The Arve catchment topography with the primary tributaries and the main cities.

**Materials and methods.**

- **Comment 5:** *l. 254: … can not be estimated by …*

It was corrected.

- **Comment 6:** *l. 255/256: "The most recent sources are often highly informative, allowing impacts to be more precisely located"*

Since 1930, there has been an increase in the number of impacts that can be localized at the scale of a building (e.g. a campground in 2009) or a neighborhood. At the same time, we see a decrease in impacts that could only be localized at the commune level.

- **Comment 7:** l. 285ff: This statement belongs to the "Outlook" section.

We agree with this and incorporated this statement to the "Outlook" section (l. 530).

**Results and discussion.**

- **Comment 8:** *Figure 2: The caption could be more informative. I assume that ticks indicate mentions, and the shaded area displays cumulative mentions, but this is somewhat speculative.*

Please find below the modified caption.

[Figure]

Figure 2. Number and diversity of studied sources to document flood impacts since 1850 in the Arve Valley. The ticks indicate the mentions and the shaded area display cumulative mentions of studied sources.

- **Comment 9:** *Figure 3: I suggest to rework this plot completely. First of all, using x-axis ticks would be helpful here. The dense number of x-axis labels does not help, since single bars are difficult to assign to specific years. Also, the second y-axis is confusing for two reasons. First, a transformation between the two y-axis of f(x) = 0.266666 * x has been used; consequently only 8 and 16 share a common y-grid line with the primary axis. All other labels float around somehow. Secondly, I do not understand why moving averages (especially for the impacts) were put on a secondary axis. Plotting data and a smoothed version of the very same data on different scales in not intuitive.*

The advantage of moving averages is that they allow to see the trend of impacts and events, which is not possible with bars. Combining the two forms of data allows us to see that in certain years peaks in the number of impacts are recorded. We can see that these years start in 1920 and increase thereafter. We choose to indicate the years with more than 25 annual recorded impacts.

If the moving averages were drawn on the left y-axis, the amplitude of the variation would be too small to be clearly visible. Hence, having two scales for the y-axis made the moving average of events makes the figure more readable.

[Figure]

Figure 3. Representation of the yearly occurrence of impacts – as well as years with more than 25 recorded impacts – and decennial moving averages of impacts and associated flood events."

- **Comment 10:** *Figure 4: You could use some alpha/transparency for plotting the location of the impacts. Also, the very colorful background makes it somewhat difficult to discern the different colors. Personally, I find simple bar plots to be easier understandable than bar plots in polar coordinates (i.e. pie charts). Also, point types to not match (circles on the map, ellipses in the legend).*

As suggested, we modified the legend, the background and the chart in order to make it more legible. We couldn't play with transparency much because it made it difficult to discern the impacts. That's why we choose a plain gray background.

[Figure]

Figure 4. Location and distribution of flood impacts caused by the Arve river and its tributaries. The category "non-attributed" corresponds to the impacts for which it was not possible to attribute a river, either because events are related to overland flows or because the source did not mention the river.

- **Comment 11:** *l.378: The authors write that "The increase in the number of impacts starting in the 1920's and well-marked from the 1960's can be explained by multiple factors such as indirect source effect, increasing flood activity and/or increasing exposure of goods and people." It is hypothesized that this is attributable to increased exposure or the evolution of data sources, but there is no proof for these statements. One core aspect I am somewhat missing in the discussion here is the completeness of the database. It has been shown that underreporting of events is likely in the time period up to the end of WWII. Has this been taken into account, or is the dataset simply assumed to be complete?*

We agree with this, these are indeed assumptions that we question in this article. We modified this statement: "*The increase in the number of impacts starting in the 1920's and well-marked from the 1960's could be explained by multiple factors. We examined three hypotheses: an increase in flood activity, an indirect source effect and an increase in exposure of goods and people.*"
Indeed, the question of completeness is always a problem concerning databases. Historical periods, such as wars, can lead to a loss of information. However, since we are interested in societal impacts, if the impacts of a flood are not reported, we considered the impact on society to be negligible since it was not recorded. This database is not intended to be a complete flood record. It apprehends the floods through the lenses of the communities' perception and values at the time it was reported. We modified the statement l. 314 to l. 317 : "*Although very few sources (e.g. the municipal archives of Sallanches) still need to be examined, most of the main sources (newspapers, existing databases and public archives) have been analyzed in order to constitute the database. We believe that this spectrum of multiple sources ensures that no event that was deemed damageable by local communities was missed. Hence, we consider that we have a comprehensive view of past flood impacts since 1850 over the whole Arve catchment.*"

- **Comment 12:** *l.381: "Therefore, the increase in impacts cannot be explained by changes in flood occurrence, at least prior to 1990." This is an interesting finding, that of course needs to be discussed. The authors provide some explanations in the following paragraphs. However, I find these paragraphs a little bit difficult to follow. I think this section needs to be reworked with a focus on clarity, e.g. by providing a table detailing the sources per period for easier comparison, or providing some sort of visual emphasis on the main lines of thought here.*

*Ultimately, upon reading the paper I am not sure where these observed trends in impacts do come from?*

As suggested, we rearranged the section 4.3. in order to make it clearer. Initial analyses of the database show an increase in the number of impacts since the 1920s (Figure 3). In this paper we investigate three hypotheses that could explain the increase in the number of impacts recorded in the database. The first one is an increase in flood activity. However, the occurrence of event responsible for impacts did not change significantly before the 1990's (Figure 3). The second hypothesis in an indirect source effect. The Figures 2 and 5 illustrate the strong emergence of the RTM. The third hypothesis is an increase in exposure, due to significant population growth.

We list our hypothesis and then present each of them.

"*The increase in the number of impacts starting in the 1920's and 1960's could be explained by multiple factors. We examined three hypotheses: an increase in flood activity, an indirect source effect and an increase in exposure of goods and people.*

*The hypothesis of an increase in flood activity cannot explain the increase in impacts at least prior to 1990 (Figure 3). In fact, the occurrence of event responsible for impacts did not change significantly before the 1990's.*

[revised manuscript text omitted]

*The three hypotheses studied are not excluding one another, but can be combined and complement each other."*

We realized a figure detailing the sources per periods. The figure illustrates clearly the strong emergence of the RTM. It would be provided as supplementary materials to complement the section 4.1.

[Figure]

Decennial histogram of the categories of sources registered in the HIFAVa database.

- **Comment 13:** *l.400: "We can assume that, floods are more likely to be reported in newspapers as when they happen in a location known by the reader." Could the authors elaborate this? I would assume that this does not affect events of a certain magnitude?*

The notion of risk is subjective and its representation varies greatly from one individual to another. We hypothesize that one is more likely to be interested in places that he/she is attached to because it is or has been his/her place of residence or leisure practices. The Arve Valley, especially Chamonix, has been a touristic and famous destination since the middle of the 18th century. Therefore, it is possible that events which take place in the upper Arve valley receive more media covered because it is supposed to attract the attention of a larger audience.

However, we made the choice to delete this passage for more clarity.

- **Comment 14:** *l.403: illustrated by Figure 5.*

The sentence was corrected accordingly.

- **Comment 15:** *Figure 5: See comments on Figure 4. The colors of the sources are even more difficult to spot here, due to the colorful background. I suggest to use a more neutral background, the important information is contained in the points, not in the elevation.*

As suggested, we modified the background in order to make it more legible.

[Figure]

Figure 5. Comparison of the distribution of sources describing the recorded flood impacts in the Arve catchment during two periods: 1850-1959 and 1960-2015.

- **Comment 16:** *l.435: "We can see that the trends are significantly the same." Apart from the fact that trends can only be significantly different, but not significantly the same (from a statistical point of view), I would like to point out here that - naturally - impacts and events show a higher correlation in the Figure from Appendix 4 than in Figure 3.*

Thank you for pointing out this blunder: "*We see that the trends of increasing impacts are comparable.*" Indeed, the representation of impacted municipalities (Appendix Figure A2) shows a higher correlation between the moving averages of impacts (impacted municipalities) and events.

- **Comment 17:** *l.440: Changes in exposure and vulnerability are only briefly (basically 1 paragraph) discussed in a qualitative way. I assume that data for providing a more detailed assessment of this line of thought is not available?*

Thank you for this suggestion, that we hope to explore in further analyses. A possibility would be for example to analyze the evolution of land use from the study of aerial photographs. However, this is another story and a complete paper could be dedicated to this subject. This paper aims to describe the HIFAVa data set and its collection.

- **Comment 18:** *Figure 6: top right: again, secondary y-axis is somewhat difficult to read, as 0:120000 is mapped to 0:7. I am not really convinced by these plots with secondary y-axis, especially since there is no natural relationship between these two data sources. The right axis can be plotted from 0:6 (spreading the range from a visual point of view) or from 0:12 (squishing the impacts in relation to the population count), but I am not sure which would be the more "correct" one. At least, I suggest to match the color of the y-axis to the line color.*

In order to make the graphs easier to read, we modified the figure to show the population curve and the recorded impacts curve on two different graphs.

[Figure]

*Figure 6. Aerial photographs allowing to visualize the evolution of the land use and the urban sprawl growth in Chamonix and Bonneville – completed by the representation of the impacts and the growth of the population from 1848 to 2011 (© IGN).*

- **Comment 19:** *l.475: "...height are only mentioned in rare cases."*

The sentence was corrected accordingly.

- **Comment 20:** *l.481: "Impacts on industrial facilities"*

The sentence was corrected accordingly.

- **Comment 21:** Figure 7: Suggest to try a mosaic plot for visualizing these data.

Thank you for your proposal to transform the graphic into a mosaic plot. Unfortunately, the distribution of impacts by river does not allow the data to be presented in this manner. Indeed, the distribution of impacts by river is not homogeneous: the Borne River represents only 2% of the total number of impacts recorded in the database, whereas the 'Tributaries' category accounts for 53% of the impacts. The representation in mosaic plot would not allow to distinguish the rivers with a few percentage of recorded impacts.

However, we reworked the figure to make it more readable: the river categories are presented in descending order from left to right and the y-axes have been inverted so that the one on the left shows the impacts categories distribution. This facilitates reading. In addition, we pointed out in the text of the article (l.480) that the Giffre and Borne rivers had proportionally very few impacts recorded compared to the Arve or the 'Tributaries' category. *"However, when analyzing the distribution of impact categories (Figure 7), it should be kept in mind that the Borne and Giffre rivers represent respectively only 2% and 7% of the impacts recorded in the database."*

[Figure]

Figure 7. Distribution of flood impacts categories according to the river types. The class "non attributed" correspond to all impact with no assigned river (e.g. overland flows).

- **Comment 22:** *l.507: The categories used for the analysis*

The sentence was corrected accordingly.

- **Comment 23:** *l.509: Please clarify what "evolution of the assigned words" means.*

We mean that it is possible that there is an evolution of the lexicon used. That is to say that the same category can be composed of several words. It is possible that the proportion of each of these words evolves with time. We have not yet analyzed this topic in more detail.

- **Comment 24:** *Figure 8: Histograms usually do not have whitespace between the bars, as they represent continuous variables. Also, I suggest to work on the colors, as impacts have a similar color as victims, and events have a similar color as protection infrastructure.*

Thank you for your comment, we modified the display of the histogram and the colors of the curves.

[Figure]

Figure 8. Decennial histogram of the evolution of the categories of impacts divided by the number of events.

- **Comment 25:** *l.515: hinders: do you mean hides? covers? conceals?*

We meant "hides". We modified the sentence.

- **Comment 26:** *l.517: what is meant by "(16 out of 28)"? I assume an increase from 16 to 28, and not the proportion 16/28 of something? Please clarify.*

We meant that from 1850 to 2015 there are 28 mentions of victims, and 16 mentions are after 1980. This means that there has been an increase in casualty mentions since 1980. We modified the sentence to be clearer. "*For instance, at the catchment scale there is a slight increase in the number of mentions of the victims category since 1980. In fact, during the studied period (1850-2015) there are 28 mentions of victims and 16 of them are recorded after 1980.*"

- **Comment 27:** *l.515-520: I do not fully understand the point the authors try to make here. The "slight increase" (which could also be considered as a not-so-slight increase of 75%) is hardly visible in the plot, because the share of the "victims" category is quite small altogether? This is a rather trivial observation, and in fact just a matter of data presentation.*

The increase is hardly noticeable because there is a strong augmentation in the total number of mentions of impacts since 1930. This increase in casualties is ultimately flooded by the overall increase in the total number of impact mentions.

**Conclusion.**

- **Comment 28:** *Again, the authors take up the hypothesis that the observed increase in impacts could be explained by exposure and evolution of sources. While I tend to agree in principle, I would like to emphasize that this is not really shown in the paper, but remains a hypothesis. Other aspects such as mitigation measures (either technical ones or soft measures such as awareness raising) are not considered. The effects of data completeness (i.e. underreporting in the earlier years of the time period), effects of exposure and land-use are not investigated in detail.*

This manuscript aims to serves as a reference for describing the HIFAVa data set and its collection, further analyses are in progress. It also allows the presentation of research hypotheses that will be more instigated later on.

---

## Author Comment (AC2)

Dear Mr. Neil Macdonald,

Thank you for the time and effort that you have dedicated to provide valuable feedback on our manuscript. We appreciate your insightful comments.

We incorporated changes to reflect most of the suggestions you provided. Here is a point-by-point response to your comments and concerns.

**Key points.**

- **Key point 1:** *You conflate the idea of flood impacts with flood frequency/occurrence (l329-333; l373-378; l532), I think you need to take care and ensure you recognize that you are not looking at flood frequency, just the impact of flood events from historical sources. It would I believe be beneficial to present the hydrological records from the valley's studied, if these are available, and then compare the relationship between discharge and time, as this would help clarify that flood frequency has not changed just the recording of flood impacts. This would then support the argument that social and cultural changes may explain the increased impacts as buildings are increasingly constructed in risky areas.*

We agree with this and we have indeed specified that it is the occurrence of events that are responsible for impacts recorded in the database (and not "flood occurrence").

Today, hydrological data are only available for the main river, the Arve. As a consequence, these data are not very representative of the impacts recorded in the database, since they are mostly caused by small torrential streams (53%), and among them, almost a third is related to glacial tributaries. The available hydrologic data do not allow to consider impacts caused by runoff or small and fast responding tributaries (torrential or flash floods).

Indeed, it is the reason why the database is relevant since it allows to give a comprehensive representation of past hydrological events in spite of the lack of instrumental data.

- **Key point 2:** *Did you consider ranking the descriptions of impact, or consider the use of indices in assessing the historical records to determine flood severity – if not this might be a future development and worth considering.*

Thank you for this suggestion. Indeed, we have started to explore this aspect. The creation of an index could make it possible to analyze and compare the 900 impacts recorded in the database. This would make it possible to compare them with each other (severity scale) but also diachronically. To do this, we need an index that allows to classify the events according to their impacts, in spite the lack of quantitative data. Indeed, we have very few measurements of flows or water heights. We do not have more quantitative data regarding the impacts e.g. precise number of flooded houses.

However, in the case of this paper it seems slightly out of scope because, as this paper aims to describe the HIFAVa data set and its collection. This aspect will be analyzed in later studies.

- **Key point 3:** *It would be advantageous to explain how you define impact early in the paper.*

Thank you for pointing this out. By "impact" we mean any disturbance caused by a flood that has been reported in the archives. It can be the flooding of a field, the cutting of a communication route (footbridge, road etc.), the destruction of a building or victims. We have added a definition in the introduction: L.70. "*The impacts recorded in the database can include any types of human goods that has been damaged by the inundation, like a field, a communication route or a building for example.*"

- **Key point 4:** *Are changing literacy rates over the timescales considered a consideration in the region? This could be easily stated and removed as a potential variable.*

This is a very interesting point. Literacy rates can have an impact on sources such as newspapers for example. However, since the sources used to complete the database are particularly varied, it is likely that this factor plays little role.
Moreover, the literacy rate is very high in French Northern Alps since the 18th century (Jean Nicolas, *La Savoie au XVIIIe s.*, 1978).

- **Key point 5:** *Is engineering information available that would provide insights into the changing nature and elevation of any flood defences in the valleys discussed, as changing exceedance thresholds of such structures may vary vulnerability of communities. A short section present (l460), but further historical consideration would be valuable.*

What is striking in the Arve watershed is that in 1850 almost all the current diking systems were already in place. From 1880 onwards, most of the dyke construction work was completed. Most of the developments carried out in the 21st century concern the construction of weirs to fight against the generalized stream incision because of the important extraction of materials in the rivers. Repairs were carried out on dikes during the 20th century, but it was only at the beginning of the 2000s that some new works (development of thresholds, raising of dikes) were carried out. Dyke rehabilitation works were carried out on the Arve River in the Bonneville area following the 1968 flood. However, since then no similar event as the one of 1968 has yet happened. Therefore, it is not possible to draw conclusions on the impact of the raising of the dykes.

- **Key point 6:** *It would be interesting to see some sense of data completeness, or assessment of data availability through time, we have previously tried this (see https://doi.org/10.5194/hess-21-1631-2017).*

We read with interest this study, thank you for recommending it.
However, this methodology seems difficult to apply to the analysis of the HIFAVa database. Indeed, the available hydrological data cover only the last few decades and concern only the main river. Moreover, the database mainly lists small-scale events and there are only a few high magnitude flood events.

- **Key point 7:** *You need to provide more detailed figure captions.*

Thank you for pointing this out, we completed captions from Figures 3, 6 and 7:

"Figure 2. Number and diversity of studied sources to document flood impacts since 1850 in the Arve Valley. The ticks indicate the mentions and the shaded area display cumulative mentions of studied sources."
"Figure 3. Representation of the yearly occurrence of impacts – as well as years with more than 25 recorded impacts – and decennial moving averages of impacts and associated flood events."
"Figure 5. Comparison of the distribution of sources describing the recorded flood impacts in the Arve catchment during two periods: 1850-1959 and 1960-2015."

"Figure 6. Aerial photographs allowing to visualize the evolution of the land use and the urban sprawl growth in Chamonix and Bonneville – completed by the representation of the impacts and the growth of the population from 1848 to 2011 (© IGN)."

"Figure 7. Distribution of flood impacts categories according to the river types. The class "non attributed" correspond to all impact with no assigned river (e.g. overland flows)."

- **Key point 8:** *You allude/suggest further analysis – such work would certainly strengthen the analysis within this manuscript.*

Thank you for pointing this out, in fact we hope to be able to communicate soon about these further analyses currently conducted. For reasons of paper length, these analyses could not be presented here. Hence, this paper aims only to describe the HIFAVa data set and its collection.

- **Key point 9:** *There are now several regional databases from across France and a national database, why are all these not condensed into a single location that would facilitate searches of historical flood information, I appreciate that they may have different aspects of focus – impact compared to water level, but does having different databases not make in more challenging for future studies? Has all the data in this database also been added to the national database? I think a statement addressing this would be beneficial in the discussion/conclusion. You note several of the databases in the Introduction.*

Indeed, the centralization of information from distinct data bases would greatly facilitate research and we all regret that such a long term and sustainable task is not supported by an ad hoc national french institution. To our knowledge, today only remarkable flooding events are consistently registered through the BDHI database (https://bdhi.developpement-durable.gouv.fr/welcome) but it doesn't address less dramatic or small-scale events that are more numerous and less documented. Over the Arve catchment, only Saint-Gervais 1892 and Grand-Bornand 1987 floods are in fact recorded in this database which doesn't allow any quantitative analysis. Nevertheless, currently there is not yet a single centralized database to upload the HIFAVa database and contribute to an exhaustive national wide database.

**Supplementary comments.**

**Introduction.**

- **Comment 1:** *l.84. You might also consider the CBHE which covers the whole UK and is one of the largest. Andrew R. Black & Frank M. Law (2004) Development and utilization of a national web-based chronology of hydrological events/Développement et utilisation sur internet d'une chronologie nationale d'événements hydrologiques, Hydrological Sciences Journal, 49:2, -246, DOI: 10.1623/hysj.49.2.237.34835*

Thank you for recommending this article. We have added this reference in the part of the article presenting the already existing databases.

- **Comment 2:** *l.126. Is this the justification for the start date? therefore comparability after this point?*

The time step of the HIFAVa database has been reduced to 165 years, the most interesting from the point of view of the wealth of the information recorded (1850-2015).

- **Comment 3:** *l.252. You have not explained what or how you define impacts - what do you consider an impact?*

Thank you for pointing this out. As we wrote in the answer to the Key point 3, we have added a definition in the introduction: L.70. "*The impacts recorded in the database can include any types of human goods that has been damaged by the inundation, like a field, a communication route or a building for example.*"

**Results and discussion.**

- **Comment 4:** *l.300. The figure caption needs to explain what the coloured sections are behind the bar graphs - number of sources used. It is difficult to compare these as the axis on each is different - consider using two consistent axis or a log axis?*

Thank you for pointing this out, we have modified the caption in order to explain the colored sections.

[Figure]

Figure 2. Number and diversity of studied sources to document flood impacts since 1850 in the Arve Valley. The ticks indicate the mentions and the shaded area display cumulative mentions of studied sources.

We have realized a figure detailing the sources per periods. The figure illustrates clearly the strong emergence of the RTM. It would be provided as supplementary materials and to complement the section 4.1.

[Figure]

Decennial histogram of the categories of sources registered in the HIFAVa database.

- **Comment 5:** l.329. *For such a statement you should look at the hydrological data, has flood frequency increased, or is it simply that greater exposure to flooding has resulted in more flood impacts?*

We agree with this and we indeed specified that it is the occurrence of events that are responsible for impacts recorded in the database (and not "flood occurrence").

Hydrological data available cover only the main river, the Arve. As a consequence, these data are not very representative of the impacts recorded in the database, since they are mostly caused by small torrential streams (53%), and among them, almost a third is related to glacial tributaries.

In this paper we investigate three hypotheses that could explain the increase in the number of impacts recorded in the database. The first one is an increase in flood activity. However, the occurrence of event responsible for impacts did not change significantly before the 1990's (Figure 3). The second hypothesis in an indirect source effect. The Figures 2 and 5 illustrate the strong emergence of the RTM. The third hypothesis is an increase in exposure, due to significant population growth.

The three hypotheses studied are not excluding one another, but can be combined and complement each other.

- **Comment 6:** *l.329. I think you are probably alluding to the increased vulnerability here but it is not explicit, if you remove this section and move to the next it would help the reader.*

Thank you for this suggestion. We have rearranged the section 4.2 in order to make it clearer. We removed the statement from the section, but we kept the first sentence l.329. "*Only the latest period (1990-2015) of increasing impacts may be partially due to a rise in occurrence of events that are responsible for impacts.*"

In order to rearrange the section 4.2., we present the spatial distribution of impacts and then we analyze changes in impacts over time and space:

"*The analysis of the spatial distribution of the flood impacts shows that they are spread over the entire catchment (Figure 3). They are, however, mainly gathered in the Arve valley around Chamonix and Bonneville (24 and 12,5% of total impacts recorded in the Arve catchment). These high numbers may*

*be due to the fact that these towns are both among the most densely populated and the closest towns to the Arve River. The impacts caused by the Arve River floods represent 33% of all recorded impacts, and its two main tributaries, the Giffre and the Borne Rivers, have only caused 8% of the recorded impacts. In fact, most impacts are due to small torrential streams (53%). Among them, almost a third are related to glacial tributaries, while these tributaries are localized only in the uppermost part of the catchment near Chamonix. For instance, small torrential tributaries such as the Arveyron, the Grépon (left bank tributary close to Chamonix) or the Bon Nant have caused alone more impacts than the Borne River itself.*

*The Arve tributaries produced disasters characterized by numerous and major flood damage. Among them, the 1987 Borne River flooding in its uppermost part washed away the municipal campsite of the village of the Grand-Bornand causing 23 casualties and heavy economic losses (Meunier, 1990). In addition, the 1892 glacial lake outburst from the Tête Rousse glacier in the Bon Nant River (which literally translated means "Good Stream") swept away the thermal bath of Saint-Gervais (Figure A1) and 33 houses causing at least 175 casualties. The glacier was drained in 2010 and is today closely monitored to avoid such a brutal and disastrous natural event (Garambois et al., 2016). All these high impacts events are due to sudden, highly-dynamic summer floods of tributaries, often aggravated by large sediment transport.*

*The analysis of the temporal distribution of the flood impacts shows a rise of recorded impacts. From 1850 to 1920, the number of impacts fluctuates and only four years are remarkable with more than 15 impacts (1852, 1878, 1895 and 1910). From 1920, years with 15 or more impacts become more frequent (1930, 1940, 1944, 1968, 1979, 1987, 1990, 1996, 1997 and 2007) and the total amount of impacts per year reaches 54 in 1996 (Figure 4). The decennial moving average of the impacts' number highlights an overall increase over the 165 years, punctuated by periods with less frequent impacts (in 1910-1923, 1950-1960 and 1975-1980).*

*The number of recorded flood events stays relatively stable between 1.5 and 3 events per year on average until 1990, then it rises up to 4.5 events per year. Therefore, the overall increase in recorded impacts seems partly disconnected to changes in flood occurrence. Only the latest period (1990-2015) of increasing impacts may be partially due to a rise in flood occurrence. In particular, the increase in flood impacts starting in the 1920's and well-marked since the 1960's, especially by repeated years with very high numbers of impacts, may be explained by other processes as discussed in the next section."*

- **Comment 7:** *l.336. More detail required.*

Agree, we have provided more detail to the figure caption: "Figure 3. Representation of the yearly occurrence of impacts – as well as years with more than 25 recorded impacts – and decennial moving averages of impacts and associated flood events."

- **Comment 8:** *l.377. You need to be careful that you do not combine flood impacts with flood events. You have evidence of flood impacts, these may be different from flood events, i.e. some events might not be recorded, or a threshold e.g. bank was not overtopped. I do think it would help if you included the annual maximum flood series for these rivers when available.*

We agree with this and we have indeed specified that it is the occurrence of events that are responsible for impacts recorded in the database. "*However, flood occurrence of events responsible of impacts recorded in the database did not change significantly before the 1990's (Figure 3)."*

Today, hydrological data are only available for the main river, the Arve and only for the last decades.

- **Comment 9:** *l.380. Why split it in 1960, just state your justification.*

These periods correspond to the two-time steps for the analysis of the chronological evolution of the impact categories recorded in the database since 1850.
We have clarified the choice for these two periods: "*During the first period, the society of the Arve watershed is a rather homogeneous agricultural society, the river is strongly exploited with extractions and the industry develops slowly. After 1960, the Arve watershed experienced a strong tourist development and a rapid demographic expansion.*"

- **Comment 10:** *l.412. Good - have you made any attempt to explore this further?*

Since 1930, there has been an increase in the number of impacts that can be localized at the scale of a building (e.g. a campground in 2009) or a neighborhood. At the same time, we see a decrease in impacts that could only be localized at the commune level. We have not yet been able to explore this further.

- **Comment 11:** *l.430. Not statistically significant - comparable would be better.*

Thank you for pointing out this blunder: "*We see that the trends of increasing impacts are comparable.*"

- **Comment 12:** *l.447. A more detailed caption is required.*

Agree, we have provided more detail to the figure caption: "Figure 6. Aerial photographs allowing to visualize the evolution of the land use and the urban sprawl growth in Chamonix and Bonneville – completed by the representation of the impacts and the growth of the population from 1848 to 2011 (© IGN)."

- **Comment 13:** *l.460. This is an important point, are you aware of other changes to flood protection works?*

In 1850, almost all the current diking systems were already in place in the Arve watershed. From 1880 onwards, most of the dyke construction work was completed. Repairs were carried out on dikes during the 20th century, but it was only at the beginning of the 2000s that some new works (development of weirs, raising of dikes) were carried out. Dyke rehabilitation works have been carried out on the Arve River in the Bonneville area following the 1968 flood. However, since the works, an event similar to the one of 1968 has not happened again. It is not possible for the moment to conclude on the impact of these works.

- **Comment 14:** *l.510. Review sentence.*

We have reviewed the sentence: "*This choice of data representation does not allow to visualize the evolution of the absolute values.*" For example, the increase of the number of victims is hardly noticeable because there is a strong augmentation in the total number of impacts mentions since

1930. This increase in casualties is ultimately flooded by the overall increase in the total number of impact mentions.

**Conclusions.**

- **Comment 15:** *l.532. As previously stated you are just looking at impacts, therefore to make such a statement you should consider hydrological data, or reframe.*

"*This rise does not seem to be related to increased flood hazard since it does not follow changes in flood occurrence of events responsible of impacts, except partially for the latest period (1990-2015).*"

- **Comment 16:** *l.538. Large sediment or high volumes of sediment?*

We meant "high volumes". We corrected the sentence.

**Additional clarifications.**
In addition to the above comments, all spelling and grammatical errors pointed out have been corrected.

---

## Author Response (AR1)

Responses to Referees 1 and 2.

**Response to Referee 1.**

Dear Referee 1,
Thank you for your thoughtful comments, the time and effort spent.
As you wondered in your introductive commentary, this manuscript first aims to describe the new HIFAVa database (its collection and structure) and potential outputs through preliminary analyses. We completed the introduction and slightly changed the title to make this clearer. For further comments, please refer to the point-by-point response to your comments and concerns below.

**Abstract.**

- **Comment 1:** *I suggest to streamline the beginning of the abstract, which is a little bit clumsy. Some examples:*
*l. 24: suggest to specify "it"*
*l. 25: the "Alps that warm at a rate twice as high in the Northern Hemisphere" - although this will probably be understood correctly by informed readers, the statement itself is somewhat unclear*

Response 1: As suggested, we streamlined the abstract in order to make it more understandable and we cleared the statement l.18.: *"Global warming is expected to exacerbate flood risk and could be more pronounced in the European Alps which are experiencing a high warming rate, likely to lead to heavier rainfall events."*

**Introduction.**

- **Comment 2:** *l.118: I suggest to be more precise as far as the "first database documenting a mountainous catchment" is concerned. There are definitely databases on historic flooding and debris flows in other countries in the European Alps, including Switzerland, Austria and Italy. These do comprise natural hazard events on a catchment level, and some have been linked to exposure and mitigation (e.g. https://doi.org/10.1016/j.crm.2021.100294 for torrential flooding; https://doi.org/10.1016/j.gloenvcha.2020.102149 for avalanches), therefore effectively covering the "interactions between social and natural dynamics engendering flood impacts".*

Response 2: Thank you for your reading recommendations that are very relevant to our study. We would like to point out that the specificity of our database is that it was developed about flood impacts identified in the archives over historical timescale (i.e. over longer time periods that the last few decades covered by instrumental data). Publications you indicated refer either to the instrumental period (1967-2017; Schlögl et al., 2020) or to other hazard than flooding (avalanche; Zheib et al., 2021). To our knowledge, there is no similar database published (at least in international journals that we can find and read as non-german/Italian speakers). We do not know the other works you mention about flooding in small Alpine catchments. Therefore, we reworded this sentence and the one from the abstract (l.120) to consider possible similar works: *"The study of this database, probably the first one documenting flood impacts over historical time scale in a mountainous catchment, ultimately aims at analyzing the interactions between social and natural dynamics engendering these impacts."*

- **Comment 3:** *The term "impact" is a core concept throughout the manuscript. I think a clear definition on what an "impact" actually comprises would be helpful (either in the introduction*

*or in section 3). This is somewhat hidden in section 3.3, I suggest to state this more clearly earlier on.*

Response 3: We referred to the description of impacts given by the IPCC (2012) l.68. We modified the sentence to clarify it : "*Here, we consider impacts accordingly to the IPCC (2012) definition as all types of outcomes for humans, society and ecosystems occurring in the aftermath of a physical phenomenon, i.e., any disturbance, damage, casualties or destruction described in the historical archives and related to a flood event.*"

**Study area.**

- **Comment 4:** *Figure 1: Please rework the elevation legend. This is a continuous scale and should be presented as such. If a discrete scale is used, intervals need to be reported and not scalar values. Which color does an elevation of say 1000 m a.s.l. correspond to? Is it green or yellow, or something in between? In addition, I would advise to use prettier breaks, and not 401 - 1401 - 2902 - 4810, which seems somewhat arbitrary. The color of the Arve river in the legend does not correspond to the color in the map.*

Response 4: We modified the figure accordingly by using a continuous scale in grey shades to ensure the legibility of the other layers of information. We also changed the color of the Arve river to make it more discernible.

[Figure]

Figure 1. The Arve catchment location, topography, main hydrological network and the studied cities.

**Materials and methods.**

- **Comment 5:** *l. 254: ... can not be estimated by ...*

Response 5: It was corrected.

- **Comment 6:** *l. 255/256: "The most recent sources are often highly informative, allowing impacts to be more precisely located"*

Response 6: We do not know/understand why the reviewer pointed out this sentence.

- **Comment 7:** l. 285ff: This statement belongs to the "Outlook" section.

Response 7: This statement has been incorporated to the "Outlook" section (l. 539).

**Results and discussion.**

- **Comment 8:** *Figure 2: The caption could be more informative. I assume that ticks indicate mentions, and the shaded area displays cumulative mentions, but this is somewhat speculative.*

Response 8: The reviewer well understands the figure. We modified the caption so that these points are well stated. Accordingly to your comment 12, we have completed the figure with a second panel detailing the sources per periods.

[Figure]

Figure 2. a) Number and b) percentage of mentions from the studied sources to document flood impacts since 1850 in the Arve Valley. In panel a, the ticks indicate each mentions of impacts and the colored areas display cumulative number of mentions.

- **Comment 9:** *Figure 3: I suggest to rework this plot completely. First of all, using x-axis ticks would be helpful here. The dense number of x-axis labels does not help, since single bars are difficult to assign to specific years. Also, the second y-axis is confusing for two reasons. First, a transformation between the two y-axis of f(x) = 0.266666 * x has been used; consequently only 8 and 16 share a common y-grid line with the primary axis. All other labels float around somehow. Secondly, I do not understand why moving averages (especially for the impacts) were put on a secondary axis. Plotting data and a smoothed version of the very same data on different scales in not intuitive.*

Response 9: We have redesigned the x-axis and homogenized the scales of the two y-axes.
The advantage of moving averages is that they allow to highlight decadal-scale changes in number of impacts and events, which is not possible with bars. Combining the two forms of data allows us to see that in certain years peaks in the number of impacts are recorded. If the moving averages were drawn on the same y-axis than the raw data, the amplitude of the curves would be too small to be clearly visible. Hence, the two scales aim to make clearer the decadal changes in occurrence of impacts and impacts events.

[Figure]

Figure 3. Yearly occurrence of impacts and decennial moving averages of impacts (red curve) and associated impact events (blue curve). Events discussed in the article are labeled with their year of occurrence.

- **Comment 10:** *Figure 4: You could use some alpha/transparency for plotting the location of the impacts. Also, the very colorful background makes it somewhat difficult to discern the different colors. Personally, I find simple bar plots to be easier understandable than bar plots in polar coordinates (i.e. pie charts). Also, point types to not match (circles on the map, ellipses in the legend).*

Response 10: We modified the legend, the background and the chart as suggested. A plain gray background has been applied so that the different colors can be discerned.

[Figure]

Figure 4. a) Location and b) distribution of flood impacts caused by the Arve River and its tributaries. The category "non-attributed" corresponds to the impacts for which it was not possible to attribute a river, either because events are related to overland flows or because the source did not mention the river.

- **Comment 11:** *l.378: The authors write that "The increase in the number of impacts starting in the 1920's and well-marked from the 1960's can be explained by multiple factors such as indirect source effect, increasing flood activity and/or increasing exposure of goods and people." It is hypothesized that this is attributable to increased exposure or the evolution of data sources, but there is no proof for these statements. One core aspect I am somewhat missing in the discussion here is the completeness of the database. It has been shown that underreporting of events is likely in the time period up to the end of WWII. Has this been taken into account, or is the dataset simply assumed to be complete?*

Response 11: We agree with this comment since these are indeed assumptions that we question in this article. Indeed, we do not have any evidence that would allow us to choose one of them and these hypotheses remain orientations to be explored by further studies.
Consequently, we have rephrased our comments in order to clearly state that these options are open and need to be studied further.
The completeness of historical database is indeed a key issue. However, to ensure a database as complete as possible, we only considered the period of the last 150 years when sources are more abundant and richer (l.200). In addition, most available sources covering this period have been studied over the last 10 years. Therefore, we trust that this database well covers the flood impacts since 1850, except maybe during the few particular years of WWII. Due to the shortness of this war, we do not expect that this would strongly change the observed decades-long trends.

- **Comment 12:** *l.381: "Therefore, the increase in impacts cannot be explained by changes in flood occurrence, at least prior to 1990." This is an interesting finding, that of course needs to be discussed. The authors provide some explanations in the following paragraphs. However, I find these paragraphs a little bit difficult to follow. I think this section needs to be reworked with a focus on clarity, e.g. by providing a table detailing the sources per period for easier comparison, or providing some sort of visual emphasis on the main lines of thought here. Ultimately, upon reading the paper I am not sure where these observed trends in impacts do come from?*

Response 12: As suggested, we rearranged the section 4.3. in order to make it clearer. Initial analyses of the database show an increase in the number of impacts since the 1920s (Figure 3).
We completed the figure detailing the sources per periods (see comment 8).

- **Comment 13:** *l.400: "We can assume that, floods are more likely to be reported in newspapers as when they happen in a location known by the reader." Could the authors elaborate this? I would assume that this does not affect events of a certain magnitude?*

Response 13: We deleted this passage since we cannot elaborate this and it was not of importance.

- **Comment 14:** *l.403: illustrated by Figure 5.*

Response 14: The sentence was corrected accordingly.

- **Comment 15:** *Figure 5: See comments on Figure 4. The colors of the sources are even more difficult to spot here, due to the colorful background. I suggest to use a more neutral background, the important information is contained in the points, not in the elevation.*

Response 15: As suggested, we modified the background in order to make it more legible.

[Figure]

Figure 5. Comparison of the spatial distribution of impacts by categories of sources in the Ave catchment during the periods (a) 1850-1959 and (b) 1960-2015.

- **Comment 16:** *l.435: "We can see that the trends are significantly the same." Apart from the fact that trends can only be significantly different, but not significantly the same (from a statistical point of view), I would like to point out here that - naturally - impacts and events show a higher correlation in the Figure from Appendix 4 than in Figure 3.*

Response 16: Thank you for pointing out this blunder: "*We see that in both figures the trends of increasing impacts are similar.*" Indeed, the representation of impacted municipalities (Appendix Figure A2) shows a higher correlation between the moving averages of impacts (impacted municipalities) and events. Certainly, because the extend of the flood events are most of the time at the scale of a municipality.

- **Comment 17:** *l.440: Changes in exposure and vulnerability are only briefly (basically 1 paragraph) discussed in a qualitative way. I assume that data for providing a more detailed assessment of this line of thought is not available?*

Response 17: Thank you for this suggestion that we hope to explore in further analyses. Indeed, the dataset required to analyze exposure and vulnerability is not available so far and still requires a long work. In addition, this goes well beyond the scope of this paper, which first aims to describe the HIFAVa data set and identify the related scientific questions for further studies.

- **Comment 18:** *Figure 6: top right: again, secondary y-axis is somewhat difficult to read, as 0:120000 is mapped to 0:7. I am not really convinced by these plots with secondary y-axis, especially since there is no natural relationship between these two data sources. The right axis can be plotted from 0:6 (spreading the range from a visual point of view) or from 0:12 (squishing the impacts in relation to the population count), but I am not sure which would be the more "correct" one. At least, I suggest to match the color of the y-axis to the line color.*

Response 18: We have homogenized the scales of the two y-axes. We consider absolute values in a secondary order since our interest is first to observe whether changes in the number of impacts and in the population growth are synchronous.

[Figure]

*Figure 6. Aerial photographs highlighting changes in land use and urban sprawl growth in Chamonix and Bonneville as well as plots stressing changes in impacts and population growth from 1848 to 2011 (© IGN, INSEE).*

- **Comment 19:** *l.475: "...height are only mentioned in rare cases."*

Response 19: The sentence was corrected accordingly.

- **Comment 20:** *l.481: "Impacts on industrial facilities"*

Response 20: The sentence was corrected accordingly.

- **Comment 21:** Figure 7: Suggest to try a mosaic plot for visualizing these data.

Response 21: Thank you for your proposal to transform the graphic into a mosaic plot. Unfortunately, the distribution of impacts by river does not allow the data to be presented as a mosaic plot. Indeed, the distribution of impacts by river is not homogeneous: the Borne River represents only 2% of the total number of impacts recorded in the database, whereas the 'Tributaries' category accounts for 53% of the impacts. The representation in mosaic plot would not allow to distinguish the rivers with a few percentages of recorded impacts.
However, we reworked the figure to make it more readable.

[Figure]

Figure 7. Distribution of flood impacts categories according to the river types. The class "non attributed" gathers the impacts that could not be assigned to a river (e.g. overland flows).

- **Comment 22:** *l.507: The categories used for the analysis*

Response 22: The sentence was corrected accordingly.

- **Comment 23:** *l.509: Please clarify what "evolution of the assigned words" means.*

Response 23: We mean that it is possible that there is an evolution of the lexicon used. That is to say that the same category can be composed of several words. It is possible that the proportion of each of these words evolves with time. We have not yet analyzed this topic in detail. We have specified this in the manuscript (l.598).

- **Comment 24:** *Figure 8: Histograms usually do not have whitespace between the bars, as they represent continuous variables. Also, I suggest to work on the colors, as impacts have a similar color as victims, and events have a similar color as protection infrastructure.*

Response 24: Thank you for your comment, we redesigned the x-axis, modified the display of the histogram and the colors of the curves.

[Figure]

Figure 8. Decennial histogram of the evolution of the categories of impacts and the number of events.

- **Comment 25:** *l.515: hinders: do you mean hides? covers? conceals?*

Response 25: We meant "hides". We modified the sentence (l.548): "*The data representation of Figure 8 does not allow to visualize the evolution of the absolute values*."

- **Comment 26:** *l.517: what is meant by "(16 out of 28)"? I assume an increase from 16 to 28, and not the proportion 16/28 of something? Please clarify.*

Response 26: We meant that from 1850 to 2015 there are 28 mentions of victims, and 18 mentions are after 1960. This means that there has been an increase in casualty mentions since 1960. We modified the sentence to be clearer (l.549). "*For instance, at the catchment scale there is an increase in the number of mentions of the victim category with 18 mentions during the period 1960-2015 compared to 10 for the period 1850-1959 (Figure 8)."*

- **Comment 27:** *l.515-520: I do not fully understand the point the authors try to make here. The "slight increase" (which could also be considered as a not-so-slight increase of 75%) is hardly visible in the plot, because the share of the "victims" category is quite small altogether? This is a rather trivial observation, and in fact just a matter of data presentation.*

Response 27: The reviewer well understands the point.  This is a trivial observation, but we feel it is important to mention it. The increase is hardly noticeable because there is a strong augmentation in the total number of mentions of impacts since 1930. This increase in casualties is ultimately flooded by the overall increase in the total number of impact mentions. We removed the term "slight".

**Conclusion.**

- **Comment 28:** *Again, the authors take up the hypothesis that the observed increase in impacts could be explained by exposure and evolution of sources. While I tend to agree in principle, I would like to emphasize that this is not really shown in the paper, but remains a hypothesis. Other aspects such as mitigation measures (either technical ones or soft measures such as awareness raising) are not considered. The effects of data completeness (i.e. underreporting in the earlier years of the time period), effects of exposure and land-use are not investigated in detail.*

Response 28: We fully agree with the reviewer. As a reminder, this manuscript first aims to describe the HIFAVa database (e.g. its collection, structure and content) and open research hypotheses that will be more instigated later on. Hence, we nuance/rework the text so that these hypotheses are just discussed and identified as research questions for further works.

**Response to Referee 2.**

Dear Pr. Neil Macdonald,

Thank you for the time and effort that you have dedicated to provide valuable feedback on our manuscript. We appreciate your insightful comments.

Here is a point-by-point response to your comments and concerns.

In addition to the above comments, all spelling and grammatical errors pointed out have been corrected.

**Key points.**

- **Key point 1:** *You conflate the idea of flood impacts with flood frequency/occurrence (l329-333; l373-378; l532), I think you need to take care and ensure you recognize that you are not looking at flood frequency, just the impact of flood events from historical sources. It would I believe be beneficial to present the hydrological records from the valley's studied, if these are available, and then compare the relationship between discharge and time, as this would help clarify that flood frequency has not changed just the recording of flood impacts. This would then support the argument that social and cultural changes may explain the increased impacts as buildings are increasingly constructed in risky areas.*

Response Key point 1: We agree with this comment and we have indeed specified that our database focuses on the occurrence of events that are responsible for impacts, i.e. "impact events" (and not "flood occurrence"). We have modified the manuscript accordingly by removing all mentions of flood occurrence or flood frequency.

The available hydrological data only concern the main river, i.e. the Arve River, which is poorly representative of the 154 streams associated to the recorded impacts. Most impacts recorded by the Arve River correspond to only 33% of all recorded impacts, while the increase in impacts recorded in 1990-2015 is mostly related to floods from ungauged small tributaries. In addition, these hydrological data cover only the last decades (50 years maximum) limiting the analysis of the observed changes in flood impacts starting in the 1920's. Lastly, only a very few floods occur along the Arve River during this instrumental period (1950-2015) strongly limiting the comparison.

- **Key point 2:** *Did you consider ranking the descriptions of impact, or consider the use of indices in assessing the historical records to determine flood severity – if not this might be a future development and worth considering.*

Response Key point 2: Thank you for this suggestion. Given the lack of quantitative data on hazard (measurements of discharge or water height) or impacts (e.g. precise number of flooded houses), indices would need to be developed. However, a key difficulty is to find indices considering the various nature of hazard in time and scale (between flash floods in small catchments and day-long flood of the main rivers) and the changing nature of impacts through time. Thereby, the development of such indices appears to be a specific work that goes well beyond the scope of this paper. However, we recognize this as an interesting perspective.

- **Key point 3:** *It would be advantageous to explain how you define impact early in the paper.*

Response Key point 3: Thank you for pointing this out. The definition of impact we use was at the early stage of the introduction (l.68) but it was maybe unclear either this definition was only given for example or this is the one we use. We modified the sentence to clarify this point as well as our definition (requested by reviewer 1) : "*Here, we consider impacts accordingly to the IPCC (2012) definition as all types of outcomes for humans, society and ecosystems occurring in the aftermath of a physical phenomenon, i.e., any disturbance, damage, casualties or destruction described in the historical archives and related to a flood event.*"

- **Key point 4:** *Are changing literacy rates over the timescales considered a consideration in the region? This could be easily stated and removed as a potential variable.*

Response Key point 4: This is a very interesting point. The literacy rate is very high in French Northern Alps since the 18th century (Jean Nicolas, *La Savoie au XVIIIe s.*, 1978).

- **Key point 5:** *Is engineering information available that would provide insights into the changing nature and elevation of any flood defences in the valleys discussed, as changing exceedance thresholds of such structures may vary vulnerability of communities. A short section present (l460), but further historical consideration would be valuable.*

Response Key point 5: At the time of writing very few archives that could provide engineering data had been searched. Information we collected about flood defenses are indeed reported l.135. What is striking in the Arve watershed is that in 1850 almost all the current diking systems were already in place. From 1880 onwards, most of the dyke construction work was completed and their nature did not significantly change after this date.
However, an in-depth study was carried out on the diking systems of the Arve and its tributary the Borne[1]. This study shows that most of the developments carried out in the 21st century concern the construction of weirs to fight against the generalized stream incision because of the important extraction of materials in the rivers. Repairs were carried out on dikes during the 20th century, but it was only at the beginning of the 2000s that some new works (development of thresholds, raising of dikes) were carried out.
This study is limited to the diking systems that protect the city of Bonneville and unfortunately, this research has not been replicated for the rest of the territory.

- **Key point 6:** *It would be interesting to see some sense of data completeness, or assessment of data availability through time, we have previously tried this (see https://doi.org/10.5194/hess-21-1631-2017).*

Response Key point 6: We read with interest this study, thank you for recommending it.
However, this methodology seems difficult to apply to the analysis of the HIFAVa database. Indeed, the available hydrological data cover only the last few decades and concern only the main river (see response to Key point 1).

- **Key point 7:** *You need to provide more detailed figure captions.*
* * *
[1] ACTHYS-Diffusion, 2017. *Etude pour la restauration des systèmes d'endiguement de l'Arve et du Borne. Fiches d'information historique (FIH) par système d'endiguement.*

Response Key point 7: Thank you for pointing this out, we completed captions from the following figures:

Figure 2 (see comment 5):
Figure 2. a) Number and b) percentage of mentions from the studied sources to document flood impacts since 1850 in the Arve Valley. In panel a, the ticks indicate the each mentions of impacts and the colored areas display cumulative number of mentions.

Figure 3: *more details required.*
Figure 3. Yearly occurrence of impacts and decennial moving averages of impacts (red curve) and associated impact events (blue curve). Events discussed in the article are labeled with their year of occurrence.

Figure 6: *a more detailed caption is required.*
Figure 6. Aerial photographs highlighting changes in land use and urban sprawl growth in Chamonix and Bonneville as well as plots stressing changes in impacts and population growth from 1848 to 2011 (© IGN, INSEE).

- **Key point 8:** *You allude/suggest further analysis – such work would certainly strengthen the analysis within this manuscript.*

Response Key point 8: The main aim of this paper is the publication of the HIFAVa dataset and the opening of associated key scientific questions for further studies. At this stage, some of the proposed analysis have just been started and need further developments before publication.

- **Key point 9:** *There are now several regional databases from across France and a national database, why are all these not condensed into a single location that would facilitate searches of historical flood information, I appreciate that they may have different aspects of focus – impact compared to water level, but does having different databases not make in more challenging for future studies? Has all the data in this database also been added to the national database? I think a statement addressing this would be beneficial in the discussion/conclusion. You note several of the databases in the Introduction.*

Response Key point 9: Indeed, the centralization of information from distinct data bases would greatly facilitate research and we all regret that such a long term and sustainable task is not supported by an ad hoc national French institution. Only "remarkable" (i.e. with very high impacts) flood events are consistently registered through the BDHI database ([https://bdhi.developpement-durable.gouv.fr/welcome](https://bdhi.developpement-durable.gouv.fr/welcome)). This is the reason why the overlap between BDHI and HIFAVa databases is so small. Only two events are identified in both (Figure 2a): Saint-Gervais 1892 and Grand-Bornand 1987 floods. As a result, the HIFAVa data do not fill the criteria to be compiled within the BDHI.
Moreover, most of the French databases on floods are built from flood events whereas ours is based on the compilation of impacts because our scientific question is focused on the risk rather than the hazard. This leads to differences in the design of the databases and thus to their interoperability. To our knowledge, there is no French database on the flood risk, i.e. which would be designed similarly to ours and, thereby, that would enable to merge them.

**Supplementary comments.**

**Introduction.**

- **Comment 1***: l.84. You might also consider the CBHE which covers the whole UK and is one of the largest. Andrew R. Black & Frank M. Law (2004) Development and utilization of a national web-based chronology of hydrological events/Développement et utilisation sur internet d'une chronologie nationale d'événements hydrologiques, Hydrological Sciences Journal, 49:2, -246, DOI: 10.1623/hysj.49.2.237.34835*

Response 1: Thank you for recommending this article. We have added this reference in the part of the article presenting the already existing databases (l.92).

- **Comment 2:** *l.126. Is this the justification for the start date? therefore comparability after this point?*

Response 2: The time period of the HIFAVa database has been focused over the last 165 years (1850-2015) because this period is the most interesting from the point of view of the wealth of information.

- **Comment 3:** *Are there any epigraphic flood marks that could be used?*

Response 3: To our knowledge there is no epigraphic flood marks in the Arve River watershed.

- **Comment 4:** *l.252. You have not explained what or how you define impacts - what do you consider an impact?*

Response 4: See answer to the Key point 3.

**Results and discussion.**

- **Comment 5:** *l.300. The figure caption needs to explain what the coloured sections are behind the bar graphs - number of sources used. It is difficult to compare these as the axis on each is different - consider using two consistent axis or a log axis?*

Response 5: Thank you for pointing this out. We modified the caption in order to explain the colored sections and arranged the y-axis to put them on the same scale.

[Figure]

Figure 2. a) Number and b) percentage of mentions from the studied sources to document flood impacts since 1850 in the Arve Valley. In panel a, the ticks indicate the each mentions of impacts and the colored areas display cumulative number of mentions.

- **Comment 6:** l.329. *For such a statement you should look at the hydrological data, has flood frequency increased, or is it simply that greater exposure to flooding has resulted in more flood impacts?*

Response 6: We agree with this and we indeed specified that it is the occurrence of impact events that are responsible for impacts recorded in the database (and not "flood occurrence"). See response to Key point 1 about the hydrological data.

- **Comment 7:** *l.329. I think you are probably alluding to the increased vulnerability here but it is not explicit, if you remove this section and move to the next it would help the reader.*

Response 7: Thank you for this suggestion. We have rearranged the section 4.2 in order to make it clearer. We modified the statement from the section l.346. "*Because hydrological data are available only for the Arve River and since the 1950s, it is not possible to determine if this increase in impact event frequency is linked to an increase in flood occurrence.*"

- **Comment 8:** *l.336. More detail required.*

Response 8: We agree and we have provided more detail to the figure caption: Figure 3. Yearly occurrence of impacts and decennial moving averages of impacts (red curve) and associated impact events (blue curve). Events discussed in the article are labeled with their year of occurrence.

- **Comment 9:** *l.377. You need to be careful that you do not combine flood impacts with flood events. You have evidence of flood impacts, these may be different from flood events, i.e. some events might not be recorded, or a threshold e.g. bank was not overtopped. I do think it would help if you included the annual maximum flood series for these rivers when available.*

Response 9: We agree with this and we have specified that it is the occurrence of impact events, i.e. events recorded in the historical sources because of the impacts they caused. We changed the statement l.331 (see answer comment 7).

- **Comment 10:** *l.380. Why split it in 1960, just state your justification.*

Response 10: We have clarified the choice for these two periods (l.398): *"In addition to the noticeable increase in impacts, this date marks a strong change in the Arve valley economy, from a rather homogeneous agricultural society to an industrial society exploiting the river bed for materials extractions. After 1960, the Arve watershed also experienced a strong tourist development and a rapid demographic expansion."*

- **Comment 11:** *l.412. Good - have you made any attempt to explore this further?*

Response 11: We explored this further (see Figure A2) and present it l.454.

- **Comment 12:** *l.430. Not statistically significant - comparable would be better.*

Response 12: Thank you for pointing out this blunder (l.451): "*We see that in both figures the trends of increasing impacts are similar.*"

- **Comment 13:** *l.447. A more detailed caption is required.*

Response 13: We provided more detail to the figure caption: Figure 6. Aerial photographs highlighting changes in land use and urban sprawl growth in Chamonix and Bonneville as well as plots stressing changes in impacts and population growth from 1848 to 2011 (© IGN, INSEE).

- **Comment 14:** *l.460. This is an important point, are you aware of other changes to flood protection works?*

Response 14: See response Key point 5.

- **Comment 15:** *l.510. Review sentence.*

Response 15: We have reviewed the sentence (l.548): "*The data representation of Figure 8 does not allow to visualize the evolution of the absolute values.*"

**Conclusions.**

- **Comment 16:** *l.532. As previously stated you are just looking at impacts, therefore to make such a statement you should consider hydrological data, or reframe.*

Response 16: We reworded the conclusion.

- **Comment 17:** *l.538. Large sediment or high volumes of sediment?*

Response 17: We meant "high volumes". We corrected the sentence (l.579).

---

## Author Response (AR2)

Eva Boisson

460, rue de la Piscine. IGE. OSUG-B.

38400 Saint-Martin-d'Hères

France

eva.boisson@univ-grenoble-alpes.fr

Saint-Martin-d'Hères,

January 25, 2022

Dr. Sven Fuchs

Privatdozent

Editor, *Natural Hazards and Earth System Sciences*

University of Natural Resources and Life Sciences.

Institute of Natural Hazards

Austria.

Dear Dr. Fuchs,

On behalf of my co-authors, I would like to thank again the two referees for their remarks on our manuscript entitled: "*Geo-historical database of flood impacts in Alpine catchments (HIFAVa database, Arve River, France, 1850 – 2015)*". We appreciated their thoughtful comments, the time and effort they spent.

We carefully considered their remarks and addressed every one of them to improve our manuscript accordingly. In addition, all spelling and technical errors pointed out have been corrected.
All modifications in the manuscript have been highlighted in yellow.

We would like to thank Referee 1 for the article of Ballesteros-Canovas et al. 2015. This is a very interesting work which calls for dendrogeomorphic techniques. However, this paper focuses on how the analysis of tree rings have been used in identifying and analyzing paleoflood events whereas our research uses historical archives inventorying damaging events through their socio-economics impacts. As those purposes are quite different, we did not see fit to include this reference in our paper.

Following you remarks we changed our argumentation (line: 97).

We hope this revised version of our manuscript will fulfill reviewers' expectations.

Thanking you in advance for your consideration, best regards,

Eva Boisson, PhD student.
Grenoble Alpes University. Institut des Géosciences de l'Environnement.